# DiffGS: Functional Gaussian Splatting Diffusion

**Junsheng Zhou**[*]      **Weiqi Zhang**[*]      **Yu-Shen Liu**[†]

School of Software, Tsinghua University, Beijing, China

{zhou-js24,zwq23}@mails.tsinghua.edu.cn    liuyushen@tsinghua.edu.cn

## Abstract

3D Gaussian Splatting (3DGS) has shown convincing performance in rendering speed and fidelity, yet the generation of Gaussian Splatting remains a challenge due to its discreteness and unstructured nature. In this work, we propose DiffGS, a general Gaussian generator based on latent diffusion models. DiffGS is a powerful and efficient 3D generative model which is capable of generating Gaussian primitives at arbitrary numbers for high-fidelity rendering with rasterization. The key insight is to represent Gaussian Splatting in a disentangled manner via three novel functions to model Gaussian probabilities, colors and transforms. Through the novel disentanglement of 3DGS, we represent the discrete and unstructured 3DGS with continuous Gaussian Splatting functions, where we then train a latent diffusion model with the target of generating these Gaussian Splatting functions both unconditionally and conditionally. Meanwhile, we introduce a discretization algorithm to extract Gaussians at arbitrary numbers from the generated functions via octree-guided sampling and optimization. We explore DiffGS for various tasks, including unconditional generation, conditional generation from text, image, and partial 3DGS, as well as Point-to-Gaussian generation. We believe that DiffGS provides a new direction for flexibly modeling and generating Gaussian Splatting. Project page: https://junshengzhou.github.io/DiffGS.

## 1 Introduction

3D content creation is a vital task in computer graphics and 3D computer vision, which shows great potential in real-world applications such as virtual reality, game design, film production, and robotics. Previous 3D generative models usually take Neural Radiance Field (NeRF) [41, 2, 62] as the representation. However, the volumetric rendering for NeRF requires considerable computational cost, leading to sluggish rendering speeds and significant memory burden. Recent advances of 3D Gaussian Splatting (3DGS) [28, 68, 23] have demonstrated its potential to serve as the next-generation 3D representation by enabling both real-time rendering and high-fidelity appearance modeling. Designing 3D generative models for 3DGS provides a scheme for real-time interaction with 3D creations.

The core challenge in generative 3DGS modeling lies in its discreteness and unstructured nature, which prevents the well-studied frameworks in structural image/voxel/video generation from transferring to directly generate 3DGS. Concurrent works [72, 19] alternatively transport Gaussians into structural voxel grids with volume generation models [11] for generating Gaussians. However, these methods lead to 1) abundant computational cost for high-resolution voxels, and 2) limited number of generated Gaussians constrained by the voxel resolutions. Certain voxelization schemes [19] also introduce information loss, making it challenging to maintain high-quality Gaussian reconstructions.

---

[*]Equal contribution. [†]Yu-Shen Liu is the corresponding author.

38th Conference on Neural Information Processing Systems (NeurIPS 2024).

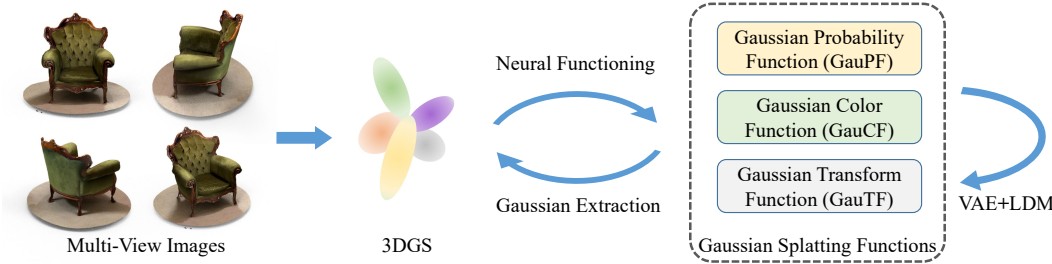

Figure 1: The illustration of DiffGS. We fit 3DGS from multi-view images and then disentangle it into three Gaussian Splatting Functions. We train a Gaussian VAE with a latent diffusion model for generating these functions, followed by a Gaussian extraction algorithm to obtain the final generated Gaussians.

To address these challenges, we present DiffGS, a novel diffusion-based generative model for general 3D Gaussian Splatting, which is capable of efficiently generating high-quality Gaussian primitives at arbitrary numbers. The key insight of DiffGS is to represent Gaussian Splatting in a disentangled manner via three novel functions: Gaussian Probability Function (GauPF), Gaussian Color Function (GauCF) and Gaussian Transform Function (GauTF). Especially, GauPF indicates the geometry of 3DGS by modeling the probabilities of each sampled 3D location to be a Gaussian location. GauCF and GauTF predict the Gaussian attributes of appearances and transformations given a 3D location as input, respectively. Through the novel disentanglement of 3DGS, we represent the discrete and unstructured 3DGS with three continuous Gaussian Splatting functions.

With the disentangled and powerful representation, the next step is to design a generative model with the target of generating these Gaussian Splatting functions. We propose a Gaussian VAE model for creating compressed representation for the Gaussian Splatting functions. The Gaussian VAE learns a regularized latent space which maps the Gaussians Splatting functions of each shape into one latent vector. A latent diffusion model (LDM) is simultaneously trained at the latent space for generating novel 3DGS shapes. With the powerful LDM, we explore DiffGS to generate diverse 3DGS both conditionally and unconditionally. Finally, we introduce a discretization algorithm to extract Gaussians at arbitrary numbers from the generated functions via octree-guided sampling and optimization. The key idea is to first extract 3D Gaussian geometry from GauPF by sampling 3D locations at the 3D spaces with the highest Gaussian probabilities, and then predict the Gaussian attributes with GauCF and GauTF. We illustrate the overview of DiffGS in Fig. 1.

We systematically summarize the superiority of DiffGS in terms of: 1) Efficiency, we design DiffGS based on Gaussian Splatting and Latent Diffusion Models, which shows significant efficiency in model training, inference and shape rendering. 2),3) Generality and quality, we generate native 3DGS without processes like voxelization, leading to unimpaired quality and generality in applying to downstream 3DGS applications. 4) Scalability, we scalably generate Gaussian primitives at arbitrary numbers. We conduct comprehensive experiments on both synthetic ShapeNet dataset and real-world DeepFashion3D dataset, which demonstrate our non-trivial improvements over the state-of-the-art methods. In summary, our contributions are given as follows.

- We propose DiffGS, a novel diffusion-based generative model for general 3D Gaussian Splatting, which is capable of efficiently generating high-quality Gaussian primitives at arbitrary numbers.

- We introduce a novel schema to represent Gaussian Splatting in a disentangled manner via three functions to model Gaussian probabilities, Gaussian colors and Gaussian transforms, respectively. We simultaneously propose a discretization algorithm to extract Gaussians from these functions via octree-guided sampling and optimization.

- DiffGS achieves remarkable performances under various tasks including unconditional generation, conditional generation from text, image, and partial 3DGS, as well as Point-to-Gaussian generation.

## 2 Related Work

### 2.1 Rendering-Guided 3D Representation

Neural implicit representations which learn signed [46, 79, 37] (unsigned [77, 78, 75]) distance functions or occupancy functions [39] have largely advanced the field of 3D generation [84, 76, 66, 36], reconstruction [80, 26, 24, 25, 45] and perception [82, 83, 81, 32, 31]. Remarkable progress have been achieved in the field of novel view synthesis (NVS) [41, 47, 62, 2, 43], with the proposal of Neural Radiance Field (NeRF) [41]. NeRF implicitly represents scene appearance and geometries using MLP-based neural networks, optimized through volume rendering to achieve outstanding NVS quality. Some subsequent variants [1, 15, 49] have shown promising performance by advancing NeRF in terms of rendering quality, scalability and view-consistency. Additionally, more recent methods [43, 7, 14, 64] explore the training and rendering efficiency of NeRF by introducing feature-grids based 3D representations. Instant-NGP [43] highly accelerates NeRF learning by introducing multi-resolution feature grids based on hash table with fully-fused CUDA kernel implementations. However, the NeRF representations which require expensive neural network inferences during volume rendering, still struggles in the applications where real-time rendering is required.

Recently, the emergence of 3D Gaussian Splatting (3DGS) [28, 59, 30, 68, 71, 18, 74] has showcased impressive real-time results in novel view synthesis (NVS). 3DGS [28] has led to revolutions in the NVS field by demonstrating superior performances in multiple domains. However, the generation of Gaussian Splatting remains a challenge due to its discreteness and unstructured nature. In this paper, we introduce a novel schema to represent the discrete and unstructured 3DGS with three continuous Gaussian Splatting Functions, thus ingeniously tackle the challenge by designing generative models for the functions.

### 2.2 3D Generative Models

The field of creating 3D contents with generative models has emerged as a particularly captivating research direction. A series of studies [48, 33, 65, 52, 40, 36, 9, 56, 69, 60] focus on optimization-based frameworks based on Score Distillation Sampling (SDS), which achieve convincing generation performances by distilling 3D geometry and appearance of the radiance fields with pretrained 2D diffusion models [44, 21] as the prior. However, these studies entail significant computational costs due to time-consuming per-scene optimization. Going beyond optimization-based 3D generation, recent methods [42, 61, 63, 27] explore 3D generative methods based on diffusion models to directly learn priors from 3D datasets for generative radiance fields modeling, which typically represent radiance fields as structural triplanes [63, 55, 17] or voxels [61, 42, 11]. DiffRF [42] leverage a voxel based NeRF representation with 3D U-Nets as the backbone to train a diffusion model.

With the recent advances in 3DGS [28], designing a powerful 3D generative model for generating 3DGS is expected to be a popular research topic. This also brings significant challenges due to the discreteness and unstructured nature of 3DGS, which prevents the well-studied frameworks in structural image/voxel/video generation from transferring to directly generate 3DGS. A series of studies [70, 86, 22, 73] follow the schema of image-based reconstruction without generative modeling, which lack the ability to generate diverse shapes. Concurrent studies GaussianCube [72] and GVGEN [19] follow the voxel-based representations to transport Gaussians into structural voxel grids with volume generation models for generating Gaussians. However, these methods come with several drawbacks, including abundant computational costs for high-resolution voxels and a restricted number of generated Gaussians constrained by voxel resolutions. Some voxelization strategies [19] may introduce information loss, leading to difficulties in preserving high-quality Gaussian reconstructions. In contrast, our proposed DiffGS explores a new perspective to directly represent the discrete and unstructured 3DGS with three continuous Gaussian Splatting Functions. Though the insight, we design a latent diffusion model for efficiently generating high-quality Gaussian primitives by learning to generate the Gaussian Splatting Functions. DiffGS generates general Gaussians at arbitrary numbers with a specially designed octree-based extraction algorithm.

## 3 Method

We introduce DiffGS, a novel diffusion-based generative model for general 3D Gaussian Splatting, which is capable of efficiently generating high-quality Gaussian primitives at arbitrary numbers.

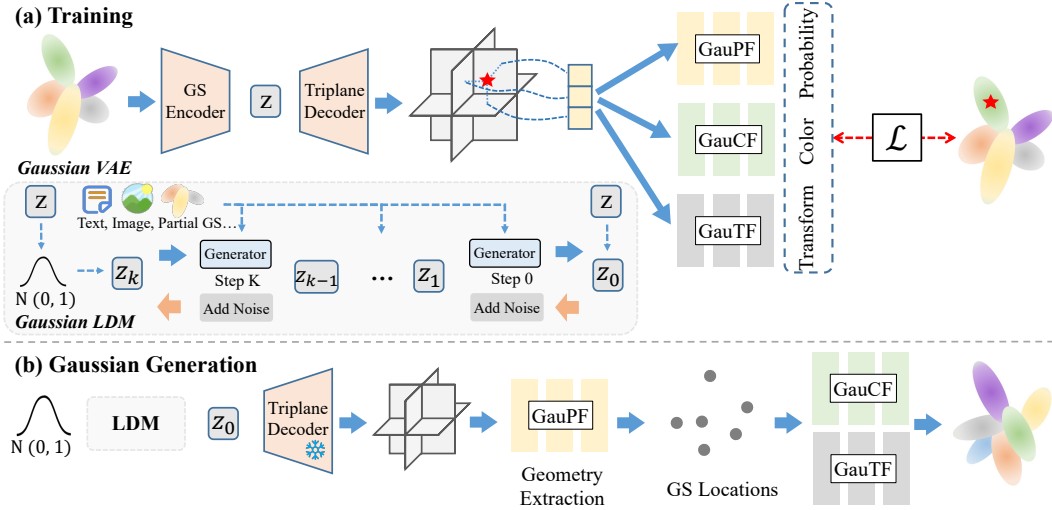

Figure 2: The overview of DiffGS. (a) We disentangle the fitted 3DGS into three Gaussian Splatting Functions to model the Gaussian probability, colors and transforms, respectively. We then train a Gaussian VAE with a conditional latent diffusion model for generating these functions. (b) During generation, we first extract Gaussian geometry from the generated GauPF, followed by the GauCF and GauTF to obtain the Gaussian attributes.

The overview of DiffGS is shown in Fig. 2. We first preview Gaussian Splatting in Sec. 3.1 and present the novel functional schema for representing Gaussian Splatting with three disentangled Gaussian Splatting Functions in Sec. 3.2. We then introduce the Gaussian Variational Auto-encoder and the Latent Diffusion Model for compressing and generative modeling on Gaussian Splatting Functions, as shown in Sec. 3.3. A novel discretization algorithm is further developed in Sec. 3.4 to extract Gaussians at arbitrary numbers from the generated functions via octree-guided sampling and optimization.

## 3.1 Preview Gaussian Splatting

3D Gaussian Splatting (3DGS) [28] represents a 3D shape or scene as a set of Gaussians with attributes to model the geometries and view-dependent appearances. For a 3DGS $G = \{g_i\}_{i=1}^N$ containing $N$ Gaussians, the geometry of i-$th$ Gaussian is explicitly parameterized via 3D covariance matrix $\Sigma_i$ and its center $\sigma_i \in \mathbb{R}^3$, fomulated as:

$$g_i(x) = \exp\left(-\frac{1}{2}(x - \sigma_i)^T \Sigma^{-1}(x - \sigma_i)\right), \tag{1}$$

where the covariance matrix $\Sigma_i = r_i s_i s_i^T r_i^T$ is factorized into a rotation matrix $r_i \in \mathbb{R}^4$ and a scale matrix $s_i \in \mathbb{R}$. The appearance of the Gaussian $g_i$ is controlled by an opacity value $o_j \in \mathbb{R}$ and a color value $c_i \in \mathbb{R}^3$. Note that the color is represented as a series of sphere harmonics coefficients in practice of 3DGS, yet we still keep its definition as three-dimension color $c_i$ in our paper for a clear understanding on our method. To this end, the 3DGS $G$ is defined as $\{g_i = \{\sigma_i, r_i, s_i, o_i, c_i\} \in \mathbb{R}^K\}_{j=1}^N$, where $K$ is dimension of the combined attributes in each Gaussian.

## 3.2 Functional Gaussian Splatting Representation

The key challenge in generative 3DGS modeling lies in its discreteness and unstructured nature, which prevents the well-studied generative frameworks from transferring to directly generate 3DGS. We address this challenge by introducing to represent Gaussian Splatting in a disentangled manner via three novel functions: Gaussian Probability Function (GauPF), Gaussian Color Function (GauCF) and Gaussian Transform Function (GauTF), respectively. Through the novel disentangling of 3DGS, we represent the discrete and unstructured 3DGS with three continuous Gaussian Splatting Functions.

**Gaussian Probability Function.** Gaussian Probability Function (GauPF) indicates the geometry of 3DGS by modeling the probabilities of each sampled 3D location to be a Gaussian location. Given a set of 3D query location $Q = \{q_j \in \mathbb{R}^3\}_{i=1}^M$ sampled in 3D space around a fitted 3DGS $G = \{g_i \in \mathbb{R}^3\}_{j=1}^N$, the GauPF of $G$ predicts the probabilities $p$ of queries $\{q_j\}_{i=1}^M$ to be a Gaussian location in $G$, fomulated as:

$$p_j = \text{GauPF}(q_j) \in [0, 1]. \tag{2}$$

The idea of Gaussian probability modeling comes from the observation that the further a 3D location $q_j$ is from all Gaussians, the lower the probability that any Gaussian occupies the space at $q_j$. Therefore the ground truth Gaussian probability of $q_j$ is defined as:

$$\text{GauPF}(q_j) = \tau(\lambda(\min_{i \in [1,N]} ||q_j - \sigma_i||_2)), \tag{3}$$

where $\min\limits_{i \in [1,N]} ||q_j - \sigma_i||_2$ indicates the distance from $q_j$ to the nearest Gaussian center in $\{\sigma_i\}_{i=1}^N$, $\lambda$ is a truncation function which filters the extremely large values and $\tau$ is a continuous function which maps the query-to-Gaussian distances to probabilities in the range of [0,1].

A learned GauPF implicitly models the locations of 3D Gaussian centers, which is the key factor for generating high-quality 3DGS. The extraction of 3DGS centers from GauPF is then achieved with our designed Gaussian extraction algorithm which will be introduced in Sec. 3.4.

**Gaussian Color and Transform Modeling.** Gaussian Color Function (GauCF) and Gaussian Transform Function (GauTF) predict the Gaussian attributes of appearances and transformations from Gaussian geometries. Specifically, given the center $\sigma_i$ of a Gaussian $g_i$ in $G$ as input, GauCF predicts the color attribute $c_i$ and GauTF predicts the rotation $r_i$, scale $s_i$ and opacity $o_i$, formulated as:

$$\{c_i\} = \text{GauCF}(\sigma_i); \quad \{r_i, s_i, o_i\} = \text{GauTF}(\sigma_i). \tag{4}$$

Note that GauCF and GauTF mainly focus on predicting the Gaussian colors and transforms from 3D Gaussian centers. This is different from the GauPF which models the probabilities of query samples in the 3D space. The reason is that GauPF focuses on exploring the geometry of 3DGS from the 3D space, while GauCF and GauTF learn to predict the Gaussian attributes from the known geometries.

Through the novel disentanglement of 3DGS, we represent the discrete and unstructured 3DGS with three continuous Gaussian Splatting Functions. The functional representation is a general and flexible term for 3DGS which has no restrictions on the Gaussian numbers, densities, geometries, etc.

## 3.3 Gaussian Variational Auto-encoder and Latent Diffusion

With the disentangled and powerful representation, the next step is to design a generative model with the target of generating these Gaussian Splatting Functions. We follow the common schema to design a Gaussian Variational Auto-encoder (VAE) [29] with a Latent Diffusion Model (LDM) [44] as the generative model. The detailed framework and the training pipeline of DiffGS are illustrated in Fig. 1(a).

**Gaussian VAE.** The Gaussian VAE compresses the Gaussian Splatting Functions into a regularized latent space by mapping the Gaussian Splatting Functions of each 3DGS shape into a latent vector, from which we can also recover the Gaussian Splatting Functions. Specifically, the Gaussian VAE consists of 1) a GS encoder $\phi_{en}$ to learn representations from 3DGS and encodes each 3DGS into a latent vector $z$, 2) a triplane decoder $\phi_{de}$ which decodes the latent $z$ into a feature triplane, and 3) three neural predictors $\psi_{pf}, \psi_{cf}$ and $\psi_{tf}$ which serve as the implementation of GauPF, GauCF and GauTF to predict Gaussian probabilities, colors and transforms, respectively.

Given a fitted 3DGS $G = \{g_i\}_{j=1}^N$ as input, the GS encoder $\phi_{en}$ extracts a global latent feature $z$ from $G$, which is then decoded into a feature triplane $t \in \mathbb{R}^{H \times W \times C \times 3}$ with the decoder $\phi_{de}$, formulated as:

$$z = \phi_{en}(G); \quad t = \phi_{de}(z). \tag{5}$$

The triplane $t$ consists of three orthogonal feature planes $\{t_{XY}, t_{XZ}, t_{YZ}\}$ which are aligned to the axex. For a 3D location $q_j$, we obtain its corresponding feature $f_j = interp(t, q_j)$ from the triplane $t$ by projecting $q_j$ onto the orthogonal feature planes and concatenating the tri-linear interpolated

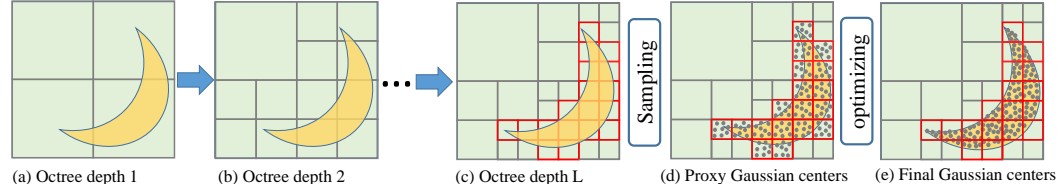

(a) Octree depth 1       (b) Octree depth 2       (c) Octree depth L       (d) Proxy Gaussian centers       (e) Final Gaussian centers

Figure 3: Gaussian geometry extractions from generated GauPF. The yellow and green regions indicate the high probability area and the low probability area judged by GauPF. (a),(b) and (c) show the progressively octree build process at depth 1,2 and $L$. (d) We sample proxy Gaussian centers from the octree at final depth $L$. (e) We optimize proxy centers to the exact geometry indicated in GauPF.

features at the three planes. We then predict the Gaussian probability, color and transform of $q_j$ with the neural Gaussian Splatting Function predictors as:

$$\{\hat{p}_i\} = \psi_{pf}(f_j); \quad \{\hat{c}_i\} = \psi_{cf}(f_j); \quad \{\hat{r}_i, \hat{s}_i, \hat{o}_i\} = \psi_{tf}(f_j). \tag{6}$$

The Gaussian VAE is trained with the target of accurately predicting Gaussian attributes and robustly regularizing the latent space. In practice, the training objective is formulated as:

$$\mathcal{L}_{\text{VAE}} = \|\{\hat{p}, \hat{c}, \hat{r}, \hat{s}, \hat{o}\} - \{p, c, r, s, o\}\|_1 + \beta \left(D_{KL}\left(\mathcal{Q}_\phi(z|G)\|\mathcal{P}(z)\right)\right). \tag{7}$$

The first loss term indicates the $\mathcal{L}_1$ loss between the predicted Gaussian attributes in Eq. (6) and the target ones defined by the ground truth Gaussian Splatting Functions in Eq. (2) and Eq. (4). The second term in Eq. (7) is the KL-divergence loss with a factor of $\beta$, which constrains on the regularization of the learned latent space of $z$. Specifically, we define the inferred posterior of $z$ as the distribution $\mathcal{Q}_\phi(z|G)$, which is regularized to align with the Gaussian distribution prior $\mathcal{P}(z) = \mathcal{N}(0, I)$, where $I$ is the standard deviation.

**Gaussian LDM.** With the trained Gaussian VAE in place, we are now able to encode any 3DGS into a compact 1D latent vector $z$. We then train a latent diffusion model (LDM) [44] efficiently on the latent space. A diffusion model is trained to generate samples from a target distribution by reversing a process that incrementally introduces noise. We define $\{z_0, z_1, ..., z_K\}$ as the forward process $\gamma(z_{0:K})$ which gradually transforms a real data $z_0$ into Gaussian noise ($z_T$) by adding noises. The backward process $\mu(z_{0:K})$ leverages a neural generator $\mu$ to denoise $z_K$ into a real data sample.

To achieve controllable generation of 3DGS, we introduce a conditioning mechanism [54] into the diffusion process with cross-attention. Given an input condition $y$ (e.g. text, image, partial 3DGS), we leverage a custom encoder $\delta$ to project $y$ into the condition embedding $\delta(y)$. The embedding is then fused into the generator $\mu$ with cross attention modules. Following DDPM, we simply adopt the optimizing objective to train the generator for predicting noises $\epsilon_\sigma$, formulated as:

$$\mathcal{L}_{\text{LDM}} = \mathbb{E}_{z_0, t, \epsilon \sim \mathcal{N}(0, I)} \left[\|\epsilon - \epsilon_\sigma\left(z_t, \delta(y), t\right)\|^2\right], \tag{8}$$

where $t$ is a time step and $\epsilon$ is a noise latent sampled from the Gaussian distribution $\mathcal{N}(0, I)$, respectively. We adopt the well-studied architecture DALLE-2 [53] as the LDM implementation.

### 3.4 Gaussian Extraction Algorithm

The final step for the generation process of DiffGS is to extract 3DGS from the generated Gaussian Splatting Functions, similar to the effect of Marching Cubes algorithm [34] which extracts meshes from Signed Distance Functions. The key factor is to extract the geometries of 3DGS, i.e., Gaussian locations and the appearances of 3DGS, i.e., colors and transforms. The full generation pipeline is shown in Fig. 2(b).

**Octree-Guided Geometry Sampling.** The locations of 3D Gaussian centers indicate the geometry of the represented 3DGS. We aim to design a discretization algorithm to obtain the discrete 3D locations from the learned continuous Gaussian Probability Function parameterized with the neural network $\psi_{pf}$, which models the probability of each query sampled in the 3D space to be a 3D Gaussian

Table 1: Comparisons of unconditional generation under ShapeNet [6] dataset.

| Method | Airplane | | Chair | |
|---|---|---|---|---|
| | FID-50K ↓ | KID-50K (‰) ↓ | FID-50K ↓ | KID-50K (‰) ↓ |
| GET3D [16] | — | — | 59.51 | 2.414 |
| DiffTF [4] | 110.8 | 9.173 | 93.02 | 6.708 |
| Ours | **47.03** | **3.436** | **35.28** | **2.148** |

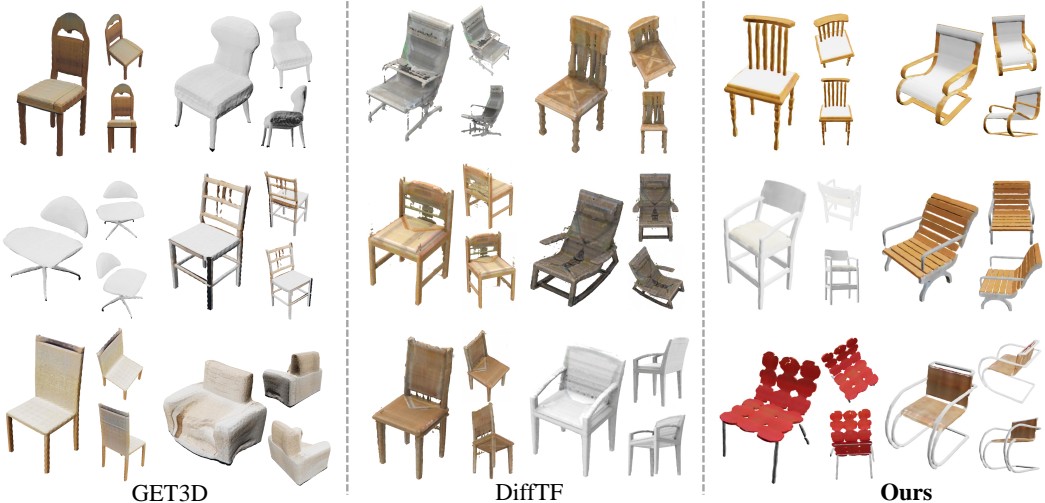

GET3D        DiffTF        **Ours**

Figure 4: Visual comparisons with state-of-the-arts on unconditional generation of ShapeNet Chairs.

location. To achieve this, we design an octree-based sampling and optimization algorithm which generates accurate center locations of 3D Gaussians at arbitrary numbers.

We show the 2D illustration of the algorithm in Fig. 3. Assume that the 3D space is divided into the high probability area (the yellow region) and the low probability area (the green region) by the generated GauPF. We aim to extract the geometry as the locations with high probabilities. A naive implementation is to densely sample queries in the 3D space and keep the ones with large probabilities as outputs. However, it will lead to high computational cost for inferencing and the discrete sampling also struggles to accurately reach the locations with largest probabilities in the continuous GauPF. We get inspiration from octree [38, 67] to design a progressive strategy which only explores the 3D regions with large probabilities in current octree depth for further subdivision in the next octree depth. After $L$ layers of octree subdivision, we reach the local regions with largest probabilities, from where we uniformly sample $N$ 3D points as the proxy points $\{\rho_i\}_{i=1}^N$ representing coarse locations of Gaussian centers.

**Optimizing Geometry with GauPF.** To further refine the proxy points to the exact locations of Gaussian centers with largest probabilities in GauPF, we propose to further optimize the proxy points with the supervision from learned GauPF $\psi_{pf}$. Specifically, we set the position of proxy points $\{\rho_i = \{\rho x_i, \rho y_i, \rho z_i\}\}_{i=1}^N$ to be learnable and optimize them to reach the positions $\{\hat{\sigma}_i\}_{i=1}^N$ with largest probabilities of $\psi_{pf}$. The optimization target is formulated as:

$$\mathcal{L}_{\text{Geo}} = -\frac{1}{N}\sum_{i=1}^N \psi_{pf}(\rho_i). \tag{9}$$

Note that we can set $N$ to arbitrary numbers, enabling DiffGS to generate 3DGS with no limits on the density and resolution.

**Extracting Gaussian Attributes.** We now obtain the estimated geometry indicating the predicted Gaussian centers $\{\hat{\sigma}_i\}_{i=1}^N$. We then extract the appearances and transforms from the generated triplane $t$, Gaussian Color Function $\psi_{cf}$ and Gaussian Transform Function $\psi_{tf}$ as

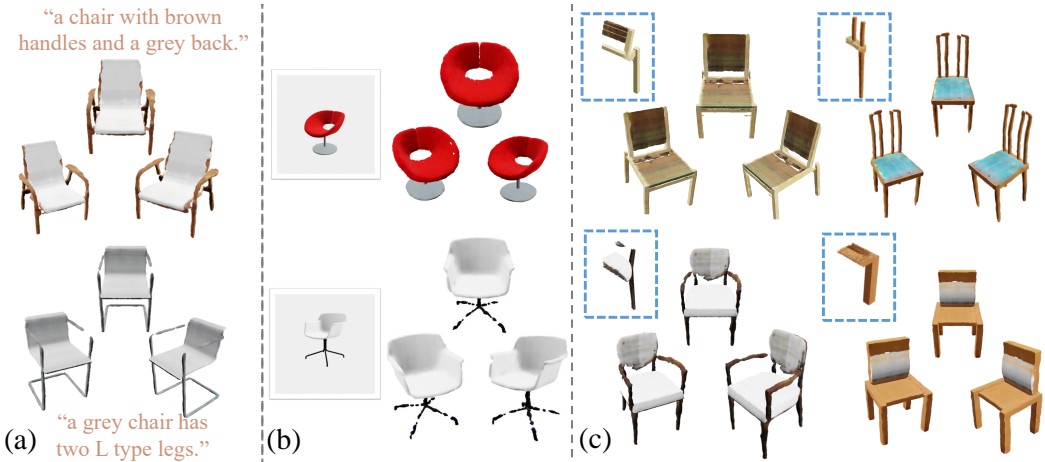

Figure 5: Visualization of conditional 3DGS generation results on ShapeNet. (a) Text conditional generation. (b) Image conditional generation. (c) Gaussian Splatting completion.

$\{\hat{c}_i\} = \psi_{cf}(interp(t, \hat{\sigma}_i))$ and $\{\hat{r}_i, \hat{s}_i, \hat{o}_i\} = \psi_{tf}(interp(t, \hat{\sigma}_i))$. Finally, the general 3DGS is now generated as $\hat{G} = \{\hat{\sigma}_i, \hat{c}_i, \hat{r}_i, \hat{s}_i, \hat{o}_i\}_{i=1}^{N}$.

## 4 Experiment

### 4.1 Unconditional Generation

**Dataset and Metrics.** For unconditional generation of 3D Gaussian Splatting, we conduct experiments under the airplane and chair classes of ShapeNet [6] dataset. Following previous works [42, 4], we report two widely-used image generation metrics Fréchet Inception Distance (FID) [20] and Kernel Inception Distance (KID) [3] for evaluating the rendering quality of our proposed DiffGS and previous state-of-the-art works. The metrics are evaluated between 50K renderings of the generated shapes and 50K renderings of the ground turth ones, both at the resolution of 1024×1024.

**Comparisons.** We compare DiffGS with the state-of-the-art methods in terms of the rendering quality of generated shapes, including the GAN-based methods GET3D [16] and the diffusion-based method DiffTF [4]. The quantitative comparison is shown in Tab. 1, where DiffGS achieves the best performance over all the baselines. We further show the visual comparison on the renderings of some generated shapes in Fig. 4, where the GAN-based GET3D struggles in generating complex shapes and the generations of DiffTF is blurry with poor textures. In contrast, DiffGS produces significantly more visual-appealing and high-fidelity generations in terms of rendering and geometry qualities.

### 4.2 Conditional Generation.

We explore the conditional generation ability of DiffGS given texts, images and partial 3DGS as the input conditions. All the experiments are conducted under the chair class of ShapeNet [6] dataset with commonly used data splits in previous methods [10, 35].

**Text/Image-conditional Gaussian Splatting Generation.** For introducing texts/images as the conditions for controllable Gaussian Splatting generation, we leverage the frozen text and image encoder from the pretrained CLIP [51] model as the implementation of custom text encoder $\gamma_{text}$ and $\gamma_{image}$ for achieving text/image embeddings. We then train DiffGS with the conditional optimization objective in Eq.(8). We show the visualization of some text/image conditional generations produced by DiffGS in Fig. 5(a) and Fig. 5(b). The results show that DiffGS accurately recovers the semantics and geometries described in the text prompts and the images, demonstrating the powerful capability of DiffGS in generating high-fidelity 3DGS from text descriptions or vision signals.

**Gaussian Splatting Completion.** Additionally, we explore an interesting task of Gaussian Splatting completion. To the best of our knowledge, we are the first to focus and introduce solutions for this task. Specifically, the Gaussian Splatting completion task is to recover the complete 3DGS from a

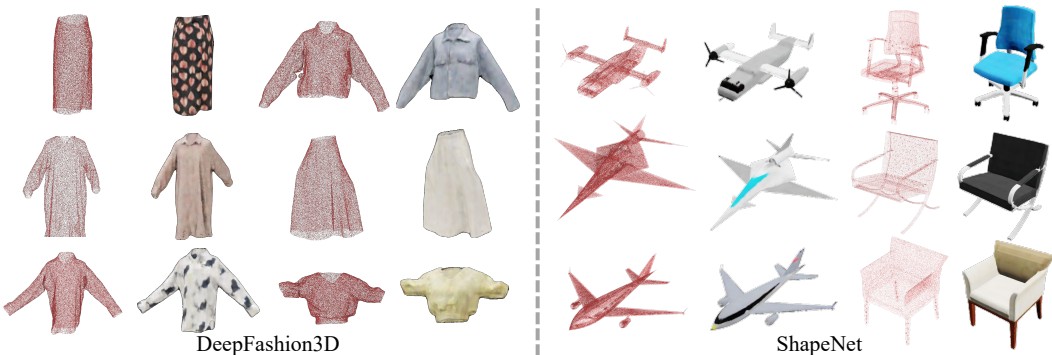

Figure 6: Visualization of Point-to-Gaussian fittings on Deepfashion3D and generations on ShapeNet.

partial 3DGS which contains large occlusions. In real-world applications, having only sparse views with limited viewpoint movement available for optimizing 3DGS often results in a partial 3DGS. Solving Gaussian Splatting completion task enables us to infer the complete and dense 3DGS from the partial ones for improving the rendering quality at invisible viewpoints.

We introduce DiffGS with partial 3DGS as the conditions for solving this task. Specifically, we simply leverage a modified PointNet [50] as the custom encoder $\gamma_{partial}$ for partial 3DGS. Fig. 5(c) presents the visualization of Gaussian Splatting completion results produced by DiffGS. The results show that DiffGS is capable of recovering complex geometries and detailed appearances from highly occluded 3DGS. Please refer to the appendix for implementation details on Gaussian Splatting completion.

### 4.3 Point-to-Gaussian Generation

We further introduce DiffGS for another challenging and vital task of Point-to-Gaussian generation. This task aims to generate the Gaussian attributes given a 3D point cloud as input. The task serves as the bridge between the easily accessible point clouds and the powerful 3DGS representation which efficiently models high-quality 3D appearances.

**Dataset and Implementation.** We conduct experiments under the chair and airplane classes of ShapeNet and also the widely-used garment dataset DeepFashion3D [85]. The DeepFashion3D dataset is a real-captured 3D dataset containing complex textures. For implementing Point-to-Gaussian, we simply train the Gaussain VAE with the three-dimension point clouds as inputs, instead of the 3DGS with attributes. Please refer to the Appendix for more details on data preparation and implementation.

**Performances.** We provide the visualization of some Point-to-Gaussian fitting and generation results in Fig. 6. We shown the fitting results for Deepfashion3D [85] dataset and the generation results for the test set of airplane and chair classes in ShapeNet [6]. DiffGS produces visual-appealing 3DGS generations given only 3D point cloud geometries as inputs. The results demonstrate that DiffGS can accurately predict Gaussian attributes for 3D point clouds. We believe DiffGS provides a new direction for 3DGS content generation by connecting 3DGS with point clouds.

### 4.4 Ablation Study

To evaluate some major designs and important hyper-parameters in DiffGS, we conduct ablation studies under the chair class of ShapeNet dataset. We report the performance in terms of PSNR, SSIM and LPIPS of the reconstructed 3DGS with Gaussian VAE.

**Framework Designs.** We first evaluate some major designs of our framework in Tab. 2. We justify the effectiveness of introducing the truncation function $\lambda$ when modeling GauPF and report the results without $\lambda$ as 'w/o truncation'. We then explore implementing the projection function $\tau$ either as $\tau(x) = e^{-x}$ (as shown in 'Exponent') or as a linear projection (as

Table 2: Ablations on framework designs.

| Method | PSNR | SSIM | LPIPS |
|---|---|---|---|
| w/o Trunction | 29.39 | 0.9792 | 0.0173 |
| Exponent | 29.74 | 0.9765 | 0.0188 |
| w/o Optimization | 30.34 | 0.9875 | 0.0152 |
| Ours | **34.01** | **0.9879** | **0.0149** |

shown in 'Ours'). We also show the results without the optimization process during Gaussian extraction as 'w/o Optimization', which demonstrates the effectiveness of optimizing Gaussians to the exact locations.

**Gaussian Numbers.** One significant advantage of DiffGS lies in the ability of generating high-quality Gaussians at arbitrary numbers. To explore how the number of Gaussians affects the rendering quality, we conduct ablations on the Gaussian numbers as shown in Fig. 3. The results demonstrate that denser Gaussians lead to better quality.

Table 3: Ablations on Gaussian number.

| Num | PSNR | SSIM | LPIPS |
|---|---|---|---|
| 50K | 28.61 | 0.9787 | 0.0251 |
| 100K | 30.35 | 0.9838 | 0.0151 |
| 350K | **34.01** | **0.9879** | **0.0149** |

## 5 Conclusion

In this paper, we introduce DiffGS for generative modeling of 3DGS. DiffGS disentangled represent 3DGS via three novel functions to model Gaussian probabilities, colors and transforms. We then train a latent diffusion model with the target of generating these functions both conditionally and unconditionally. DiffGS generates 3DGS with arbitrary numbers by an octree-guided extraction algorithm. The experimental results on various tasks demonstrate the superiority of DiffGS.

## 6 Acknowledgement

This work was supported by National Key R&D Program of China (2022YFC3800600), the National Natural Science Foundation of China (62272263, 62072268), and in part by Tsinghua-Kuaishou Institute of Future Media Data.

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

# Appendix

## A   More Experimental Details

### A.1   Gaussian Splatting Data Preparing

DiffGS takes the fitted 3DGS as input for learning generative modeling. To prepare the 3DGS dataset of ShapeNet, we uniformly render 100 views from the ground truth meshes with blender to first obtain the dense multi-view images for each 3D shape in the chair and airplane classes of ShapeNet dataset. After that, we leverage the vanilla 3D Gaussian Splatting [28] method for fitting the 3DGS for each shape with the rendered multi-view images.

For achieving more stable and regularized 3DGS data for better generative modeling, we design some strategies for better initialization and optimization of 3DGS. (1) Since we fit 3DGS from existing 3D datasets with known geometries, we can simply sample dense point clouds uniformly from the surfaces as the perfect initialization for 3DGS optimization, instead of initializing with COLMAP points. The sampled point number is set to 100K. (2) We observe that optimizing 3DGS freely may often lead to some extremly large Gaussians. This will lead to unstable training of the Gaussian VAE and latent diffusion models, further affecting the generative modeling results. Therefore, we clip the scales at a maximum size of 0.01 to avoid the abnormal Gaussians.

### A.2   Gaussian Splatting Completion

We explore the task of Gaussian Splatting completion. Specifically, the Gaussian Splatting completion task is to recover the complete 3DGS from a partial 3DGS which contains large occlusions. In real-world applications, having only sparse views with limited viewpoint movement available for optimizing 3DGS often results in a partial 3DGS. Solving Gaussian Splatting completion task enables us to infer the complete and dense 3DGS from the partial ones for improving the rendering quality at invisible viewpoints.

**Data preparation.** We generate partial 3D Gaussian Splatting data from the complete datasets in a straightforward manner. First, we randomly divide each 3DGS into 8 chunks. Then, we occlude 7 chunks, leaving the remaining chunk as the partial 3DGS. This method allows us to prepare partial-complete 3DGS pairs for training and testing.

**Implementation.** To leverage DiffGS with partial 3DGS as the conditions for Gaussian Splatting completion, we leverage a modified PointNet [50] as the custom encoder $\gamma_{partial}$ for partial 3DGS, which projects the partial 3DGS with $K$ channels into a global partial 3DGS embedding. The DiffGS for Gaussian Splatting completion is trained with the target of Eq.(8) by introducing partial 3DGS embeddings through the cross-attention module.

### A.3   Point-to-Gaussian Generation

**Data preparation.** For the task of Point-to-Gaussian generation, we first prepare the training/testing data as point cloud-3DGS pairs obtained through the Gaussian Splatting Fitting process described in Sec. A.1. Specifically, the point clouds are generated by densely sampling 100K points on the ground truth meshes, while the paired 3DGS are obtained by optimizing with multi-view images rendered around the ground truth meshes. The data preparation for the DeepFashion3D [85] dataset follows the same process as for the ShapeNet [6] dataset.

Note that we train the Point-to-Gaussian DiffGS models for fitting the DeepFashion3D dataset, which contains only 563 garment instances. In contrast, we split the airplane and chair classes of the ShapeNet dataset into train/test sets to learn generalizable representations that enables DiffGS to predict novel appearances for unknown point cloud geometries. The results shown in Fig. 6 of the main paper include both the fitting results on the DeepFashion3D dataset and the generation results from the point clouds in the test set of the airplane and chair classes in the ShapeNet dataset.

**Implementation.** For implementing Point-to-Gaussian, we train the Gaussian VAE using three-dimensional point clouds as inputs instead of 3DGS with attributes. Specifically, we replace the GS encoder in the Gaussian VAE with a PointNet-based network to learn representations from three-dimensional point clouds and recover Gaussian attributes from them. All architectures, optimization

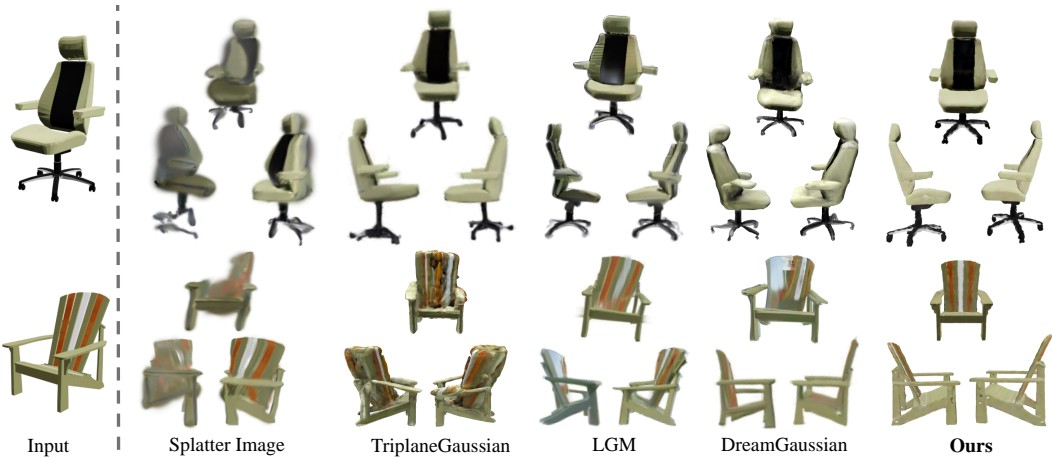

| Input | Splatter Image | TriplaneGaussian | LGM | DreamGaussian | **Ours** |

Figure 7: Qualitative comparison of image-to-3D generation.

targets, and Gaussian extraction processes remain the same as in the Gaussian VAE, except for the encoder. Note that for the Point-to-Gaussian task, the Gaussian LDM is not trained, as we focus solely on decoding and extracting Gaussians from the point cloud inputs.

# B    More Comparisons and Ablations

We further conducted comprehensive experiments focusing on two critical tasks: text-to-3D generation and single-view 3D generation. These tasks are central to demonstrating the flexibility and robustness of our approach in different application scenarios.

## B.1    Image-to-3D Generation

We compare DiffGS with various SOTA methods on implicit generation or Gaussian Splatting generation. The visual comparison of image-to-3D generation is presented in Fig. 7. These results illustrate the superior visual quality and fidelity achieved by our method compared to the SOTA baseline methods including SplatterImage [57], TriplaneGaussian [86], LGM [58] and DreamGaussian [59]. Our approach consistently produced more detailed and accurate generations, effectively capturing intricate textures and geometries that are often challenging for the compared optimization-based and multi-view based methods.

## B.2    Text-to-3D Generation

We conduct evaluations under the difficult task of text-to-3D generation. We compare DiffGS with the SOTA data-driven and optimization-based methods Shap·E [27], LGM [58] and DreamGaussian [59]. We present the visual comparisons in Fig. 8, where DiffGS achieves more visual-appealing results compared to the baselines.

We also follow the common setting to conduct quantitative comparisons using the CLIP score, a metric that measures the semantic alignment between the generated 3D models and the input conditions. The results are presented in Tab. 4. According to Tab. 4, our method achieved higher CLIP scores than both DreamGaussian and LGM. This indicates that our approach more faithfully adheres to the condition inputs.

However, it is important to note that DiffGS is trained on the ShapeNet dataset, while some methods (e.g., LGM, TriplaneGaussian) are trained on the Objaverse [13] dataset. As a result, the comparisons may not fully reflect the generation capabilities due to different data sources. The above comparisons are provided for reference, and we plan to conduct further training using the same dataset for a more accurate comparison.

A chair with a cloth seat and white legs

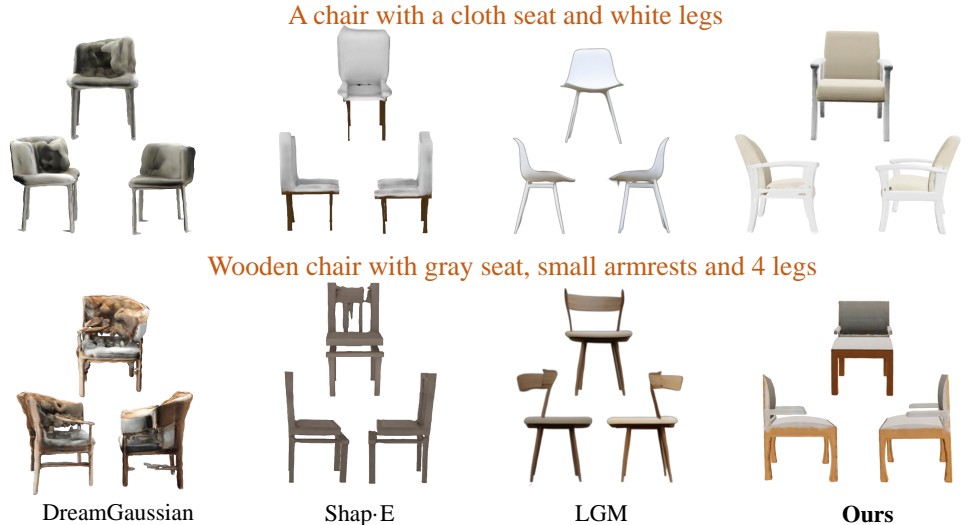

Wooden chair with gray seat, small armrests and 4 legs

| DreamGaussian | Shap·E | LGM | **Ours** |

Figure 8: Qualitative comparison of text-to-3D generation.

Table 4: Comparisons of text consistencies.

|  | DreamGaussian | Shap·E | LGM | **Ours** |
|---|---|---|---|---|
| **CLIP Scores** | 29.08 | 32.76 | 32.52 | **33.42** |

## B.3 Unconditional Generation

We compare DiffGS with DiffTF [4] on the airplane class of the ShapeNet dataset. The visual comparison is shown in Fig. 9, where our proposed DiffGS achieves more visually appealing results than DiffTF. DiffTF often produces shape generations with blurry textures, resulting in poor rendering quality. In contrast, our method produces high-fidelity renderings with the generated high-quality 3DGS, accurately capturing both geometry and appearances.

We further make a comparison with the GAN-based 3D generative model EG3D [5] and the NeRF-based SSD-NeRF [8] on unconditional generation of car models under ShapeNet dataset. The visual comparison is shown in Fig. 10, where DiffGS significantly outperforms EG3D and SSD-NeRF on the geometry and appearance details.

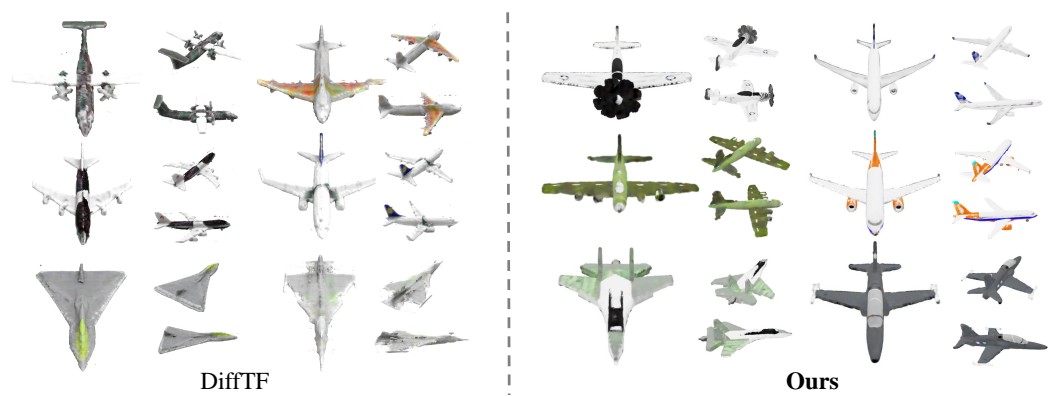

DiffTF      **Ours**

Figure 9: Visual comparisons with state-of-the-art method DiffTF [4] on unconditional generation of ShapeNet airplanes.

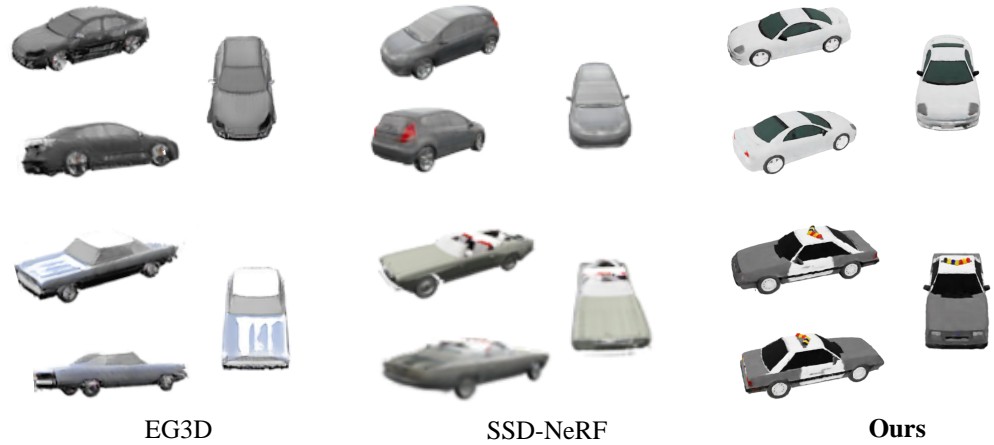

| EG3D | SSD-NeRF | **Ours** |

Figure 10: Visual comparisons with state-of-the-arts on unconditional generation of ShapeNet cars.

## B.4 Ablation Study on Octree Depth

The depth of the octree in our Gaussian extraction algorithm is an important hyper-parameter in the framework. To explore how the octree depth affects the rendering quality, we conduct ablations on the octree depth as shown in Tab. 5. The results demonstrate that a larger octree depth leads to better quality by capturing more geometry details.

The deeper octree depths may lead to increased complexity in Gaussian extraction. To address the trade-off between the efficiency and quality, we conducted an ablation study in Tab. 5 to identify the optimal balance between computational cost and reconstruction quality. The results indicate that a moderate increase in octree depths can significantly improve the Gaussian quality with very few additional cost.

Table 5: Ablations on the sample time and Gaussian quality with different octree depths.

| Depth | PSNR | SSIM | LPIPS | Sample Time (s) |
|-------|-------|--------|--------|-----------------|
| 7 | 29.77 | 0.9772 | 0.0254 | **0.15** |
| 8 | 31.70 | 0.9824 | 0.0156 | 0.16 |
| 9 | 32.70 | 0.9842 | 0.0162 | 0.21 |
| 10 | **34.01** | **0.9879** | **0.0149** | 0.58 |

## B.5 Ablation Study on Framework Designs

We also conduct ablations to demonstrate the effectiveness of introducing triplanes as the decoder implementation of our Gaussian VAE. We replace the Triplane decoder with simple Multi-Layer Perceptron (MLP) model and show the performance in Fig. 11. The results indicate that using MLP fails to capture the intricacies of Gaussian modeling adequately. The Triplane model demonstrates superior performance in terms of detail accuracy and geometric fidelity.

## C  Efficiency Analysis and Comparison

### C.1 Parameters and Inference Time

We compare the model sizes and generation time between DiffGS and other SOTA generative models. The results are shown in Tab. 6, where we highlight that DiffGS demonstrates significant efficiency compared to the SOTA baselines DiffTF, Shap·E, SSDNeRF and DreamGaussian. DiffGS also offers competitive generation time with GET3D and LGM. DiffGS has a significantly smaller number of parameters compared to most baseline methods. The smaller parameter count of DiffGS translates

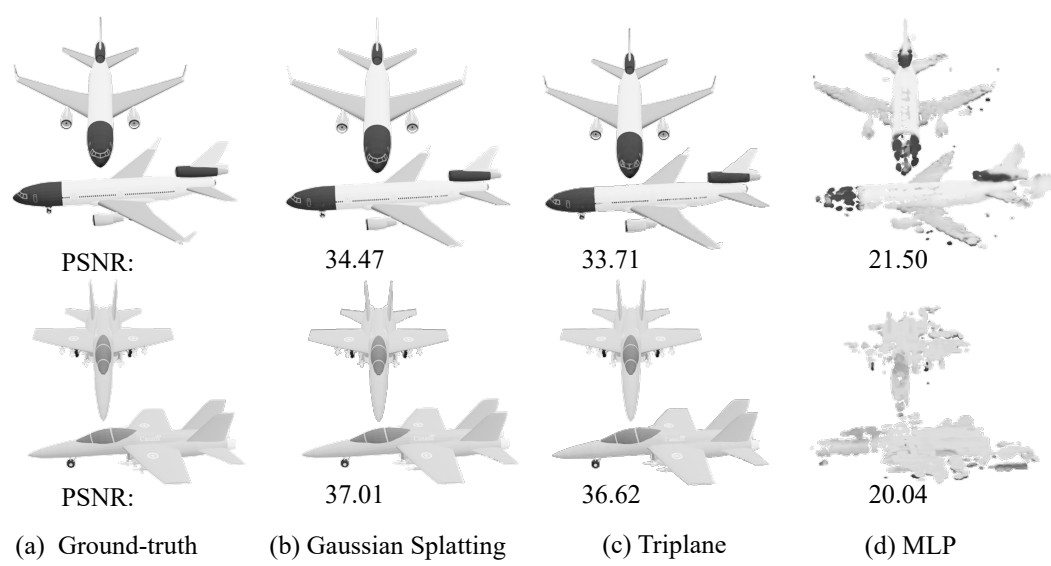

|  | PSNR: | 34.47 | 33.71 | 21.50 |
|  | PSNR: | 37.01 | 36.62 | 20.04 |
| (a) Ground-truth | (b) Gaussian Splatting | (c) Triplane | (d) MLP |

Figure 11: Ablations on 3D representation implementations. (a) The input shapes from ShapeNet. (b) Gaussians trained on the ground truth shape. (c) Reconstructed Gaussian primitives obtained with our Triplane-based Gaussian Variational Auto-encoder using proposed Gaussian Splatting functions. (d) Replace the Triplane architectures with simple MLPs.

Table 6: Comparison of model parameters and inference time. Inference time is measured on a single NVIDIA RTX 3090 GPU.

|  | GET3D | DiffTF | SSDNerf | Shap·E | DreamGaussian | LGM | **Ours** |
|---|---|---|---|---|---|---|---|
| **Parameters** (M) | **34.3** | 929.9 | 244.9 | 759.5 | 258.7 | 429.8 | 127.4 |
| **Inference Time** (s) | **5.0** | 99.7 | 27.4 | 27.1 | 197 | 10.5 | 9.5 |

into reduced memory usage and potentially faster inference times, which can be advantageous in environments where computational resources are limited.

## C.2 Gaussian Numbers during Extraction

We conducted supplementary experiments to analyze the impact of the number of extracted Gaussian points on optimization time. The results of this ablation study are presented in Tab. 7. The results show that for generations with a smaller number of Gaussians, e.g., 50K, the optimization is extremely fast, taking only 0.64 seconds to converge. For high-quality Gaussian generations with 350K primitives, the optimization time increases to 2.5 seconds, which is still efficient. In our experiments, we selected a configuration of 350K Gaussian points. This choice balances quality and computational efficiency, providing a robust representation of the model without excessively increasing processing time.

Table 7: Ablations on Gaussian Number. Optimization time is measured on a single NVIDIA RTX 3090 GPU.

| Num | PSNR | SSIM | LPIPS | Opt. Time (s) |
|---|---|---|---|---|
| 50K | 28.61 | 0.9787 | 0.0251 | **0.64** |
| 100K | 30.35 | 0.9838 | 0.0151 | 1.26 |
| 350K | **34.01** | **0.9879** | **0.0149** | 2.5 |

## D  Implement Details

We implement DiffGS with Pytorch Lightning. We leverage the Adam optimizer with a learning rate of 0.0001. We train DiffGS with eight 3090 GPUs and the convergence in each class of ShapeNet dataset takes around 5 days. The Guassian encoder and Triplane decoder are implemented based on the SDF-VAE encoder and decoder of DiffusionSDF [12], where we modify the PointNet [50] in the SDF-VAE encoder to receive $K$ dimension 3DGS as the inputs.

## E  Limitation

One limitation of our method is that it sometimes produces overly creative color schemes. We illustrate this issue with two examples in Fig. 12. As shown, the failure cases of DiffGS result in excessively colorful appearances. For instance, the chair shown exhibits a color transition from yellow to blue to green and finally to orange. These overly creative generations may not accurately reflect real-world shapes.

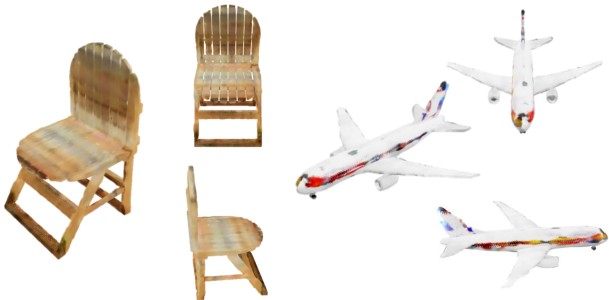

Figure 12: Failure cases of our method. DiffGS sometimes generates overly creative shapes with colorful appearances.

