# OpenReview forum: "DiffGS: Functional Gaussian Splatting Diffusion"
_NeurIPS.cc/2024/Conference — NeurIPS 2024 poster_

### Official Review · Reviewer_TmLe · 2024-06-27

**Soundness:** 2
**Presentation:** 1
**Contribution:** 2
**Rating:** 4
**Confidence:** 4

**Summary:**

This paper presents a method to turn the representation of the gaussian splatting as a point cloud into a VAE representation. With that representation it is shown how to perform  several tasks such as unconditional generation, conditional generation, and point-to-gaussian generation, all of which are done using a diffusion model that operates on the latent . It is shown that it performed better than previous methods on these tasks.

**Strengths:**

The method presented in the paper is a novel way to incorporate the gaussian splatting attributes into generation tasks. The method shows better results on some  tasks than previous methods.

**Weaknesses:**

The main one is that while reading the paper I did not get the answer to the question why enforcing a VAE to predict gaussian splatting should work better than a simpler method. It was shown that this is the case by experiments, but what is the motivation? As SOTA NeRF methods produce better reconstruction than the vanilla GS. Since the paper presents a method to create a new 3d object from the gaussian splatting primitives the relevant papers from the early (1990-2002) use of  gaussian splatting should be referred to. Additionally the number of quantitative results is limited (even in comparison to the paper of the referred baselines), and makes it hard to evaluate the method. Moreover, the second part of the first sentence in the abstract “... yet the generation of Gaussian Splatting remains a challenge due to its discreteness and unstructural nature.” is IMHO logically challenging, only after reading the introduction it becomes clear that the authors referred to generation novel 3d assets using gaussian splatting. The main concepts and ideas in the paper are interesting, but there is still work required in both writeup and experimental parts.

**Questions:**

see weaknesses.

**Limitations:**

see weaknesses.

---

> ### Author Rebuttal · Authors · 2024-08-07
>
> It is with great appreciation that we acknowledge the insightful feedback from Reviewer TmLe. We have addressed the questions and comments with careful consideration, and we encourage continued conversation to ensure the robustness of our findings.
>
> **Q1: Motivation, Literature, and Experimental Evaluation**
>
> Variational Autoencoders (VAEs) are well-suited for capturing complex distributions in a lower-dimensional latent space, which encodes diverse data into a compact latent representation, allowing for more efficient and structured generation of 3D models. In comparison to Neural Radiance Fields (NeRF), 3D Gaussian Splatting presents multiple benefits, especially regarding rendering efficiency and superior visual fidelity. Addressing the challenge of 3DGS generation is both timely and essential for the progression of the 3D modeling domain. The revised abstract will explicitly state the novelty and motivation behind using Gaussian splatting for novel 3D asset generation, emphasizing its significance in overcoming the limitations of discreteness and structural complexity.
>
> We carried out further experiments to compare our approach with the state-of-the-art baseline methods, such as EG3D (CVPR 2022), SSDNeRF (ICCV 2023), Shap·E, SplatterImage (CVPR 2024), TriplaneGaussian (CVPR 2024), LGM (ECCV 2024), and DreamGaussian (ICLR 2024). As shown in Figure A, Figure B and Figure D of the rebuttal PDF, our method shows more refined geometry and detailed textures. Besides, we compare text consistencies with the other SOTA methods in Table A of the rebuttal PDF. Our method demonstrates a competitive alignment between the text descriptions and the resulting 3D models. Besides, we provide a comparison of model parameter counts and generation times in Table B of the rebuttal PDF. DiffGS has significantly fewer parameters than LGM, resulting in reduced memory usage and computational overhead. Moreover, our approach achieves faster generation times than other state-of-the-art 3D generative models, demonstrating its efficiency in creating high-quality 3D content.

---

> > ### Comment · Reviewer_TmLe · 2024-08-13
> > **Thanks for the answers. We'll keep the rating.**
> >
> > More efforts and motivation is required.

---

### Official Review · Reviewer_cAhv · 2024-07-01

**Soundness:** 3
**Presentation:** 3
**Contribution:** 3
**Rating:** 7
**Confidence:** 4

**Summary:**

This paper proposes DiffGS, a novel diffusion-based generative model that can generate 3D Gaussian Splatting (3DGS) representing an object. As 3DGS is naturally discrete and unstructured, it is not practical to directly diffuse the 3DGS representation. Thus, the authors propose to reconstruct the 3DGS with structured triplanes that represent three functions: a probability function that models the geometry (the mean of Gaussians) of 3DGS, a color function that models the color, and a transform function that models the transformation parameters of the Gaussians. Additionally, the authors propose a VAE that encodes 3DGS into a latent vector. By representing 3DGS with such structured data, it is then possible to train a diffusion model on the latent vectors, which can be recovered to 3DGS for rendering. Experiments show that the proposed DiffGS achieves preferable quality results compared to baselines and by adding conditions to the diffusion model, it can achieve various applications.

**Strengths:**

1. The paper is well-written and easy to follow.
2. The idea of representing 3DGS with disentangled functions is novel and practical. Indeed, it is difficult to diffuse unstructured data like 3DGS, and representing 3DGS using functions that can both encode and decode is a novel direction.

**Weaknesses:**

1. It can be observed that there might be holes and incomplete shapes in the generated 3DGS (Fig.5(b)). I suspect the reason for this is that the proposed Gaussian extraction algorithm struggles to sample points in high-frequency parts like the legs of chairs. While this can be improved by increasing the number of sampled Gaussians or the depth of the octree, it might be quite computationally intensive.
2. The inference efficiency is not shown in the experiments. During each sample, DiffGS requires optimization over the sampling points, which may further slow down the inference speed of the model. The authors should provide the influence of the optimization on the inference time. However, it is acceptable if the inference speed is slower than baselines since diffusion models are naturally slow to inference.
3. An important baseline is missing. Splatter Image [1] is also a 3DGS-based object reconstruction method and shares the similar idea of representing unstructured 3DGS with structured data. Since the Splatter Image models trained on ShapeNet are also available, it is reasonable to compare DiffGS with Splatter Image under the single-view reconstruction task.
4. The authors do not provide the number of parameters of DiffGS and the comparisons with baselines. In the quality comparisons, the authors should keep DiffGS and other baselines having around the same number of parameters.

[1] Splatter Image: Ultra-Fast Single-View 3D Reconstruction, CVPR 2024

**Questions:**

1. What is the inference time of DiffGS, and how much does the optimization process during inference influence the inference speed?
2. How many parameters does the model use?

**Limitations:**

Yes, the limitations are discussed in the paper.

---

> ### Author Rebuttal · Authors · 2024-08-07
>
> We extend our sincere thanks to Reviewer cAhv for their comprehensive review and valuable suggestions. In response to the concerns raised, we have offered detailed explanations in the subsequent section, and we are eager to engage in further dialogue.
>
> **Q1: Incomplete Shapes in 3DGS**
>
> It is indeed possible that some models exhibit holes and incomplete shapes, especially in regions with high-frequency details. This issue arises from challenges in capturing fine details during the Gaussian extraction process. One straightforward approach to improve this is to increase the number of sampled Gaussians. By doing so, we can achieve a higher resolution representation, capturing more details in the high-frequency regions. Similarly, increasing the depth of the octree can enhance the model's ability to resolve fine details. We aim to balance quality and computational efficiency. While increasing the number of Gaussians and octree depth improves detail capture, it also escalates computational costs. We focus on a balance between satisfactory results and inference efficiency,, and find it acceptable to occasionally generate shapes with minor artifacts.
>
> **Q2: Inference Efficiency and Optimization Impact**
>
> During Gaussian extraction, DiffGS performs optimization over the Gaussian positions to enhance the quality and accuracy of the generated 3D Gaussian Splatting geometries. This optimization step, while essential for ensuring high-quality outputs, does introduce additional computational overhead. We conducted supplementary experiments to analyze the impact of the number of Gaussian points on optimization time. The results of this ablation study are presented in Table C of the rebuttal PDF. In our experiments, we selected a configuration of 350K Gaussian points. This choice balances quality and computational efficiency, providing a robust representation of the model without excessively increasing processing time. The optimization time for this configuration is approximately 2.5 seconds. We also compared our model's generation time with other state-of-the-art 3D generative methods. The results are shown in Table B of the rebuttal PDF, where we highlight that our method demonstrates significant efficiency compared to the SOTA baselines DiffTF, ShapE, SSDNeRF and DreamGaussian,  despite the added optimization step. DiffGS also offers competitive generation time with GET3D and LGM.
>
> **Q3: Comparison with Splatter Image Method**
>
> For a comprehensive comparison with Splatter Image, we evaluated both methods under the single-view generation task. We conducted supplementary experiments to compare the visual quality of reconstructions generated by DiffGS and Splatter Image. The results are presented in Figure A of the rebuttal PDF. The visual comparison indicates that DiffGS produces higher fidelity reconstructions with better handling of fine details and fewer artifacts compared to Splatter Image.
>
> **Q4: Number of Parameters and Baseline Comparisons**
>
> We have included the number of parameters for DiffGS and other baseline methods in Table B of the rebuttal PDF. As shown in the table, DiffGS has a significantly smaller number of parameters compared to most baseline methods. The smaller parameter count of DiffGS translates into reduced memory usage and potentially faster inference times, which can be advantageous in environments where computational resources are limited. Despite having fewer parameters, DiffGS demonstrate superior performance, demonstrating that its architecture efficiently captures essential features for high-quality 3D reconstruction.

---

### Official Review · Reviewer_6WmW · 2024-07-12

**Soundness:** 2
**Presentation:** 3
**Contribution:** 3
**Rating:** 5
**Confidence:** 4

**Summary:**

The paper is about generating 3D objects by using gaussian splatting. The method converts the points (with gaussian properties) to continuous fields. With this idea, the irregular structured data can be easily processed by neural networks.

**Strengths:**

The idea is interesting and novel. Usually we need many points (10k-1M) to represent a single object. With this large point clouds, it is very challenging to process the data with neural networks. Thus the paper proposed to convert a point cloud to a field which can be easily processed by neural networks.

The octree design seems to be also interesting. This is useful to get fine-grained structure.

**Weaknesses:**

The results shown in the supplementary video is not very satisfying. Even the paper showed some metrics comparing to some existing methods, the visual quality is much worse.

Specifically, the metrics shown in Table 1 are better than some prior works. However, I believe some important works are missing in the table, e.g., EG3D, DiffRF, SSDNerf. The evaluation protocol seems to be different from these works. I am curious how the authors pick 50k images as the reference set. SSDNerf used testing set as the reference set. I know there is always a debate between training and testing set in the evolution of generative models. But this should be mentioned in the paper.

For the visual quality, results of EG3D and DiffRF are much better than this paper. It would be better the authors can do a visual comparison with these works.

**Questions:**

How do the authors optimize the loss in Eq 6?
I understand how the point are converted to fields thus we can query the probabilities continuously in the space
But how about other properties? They are not converted to continuous fields.

**Limitations:**

Yes

---

> ### Author Rebuttal · Authors · 2024-08-07
>
> We are truly grateful for the insightful review provided by Reviewer 6WmW. Your thorough analysis and constructive feedback have significantly enhanced the quality of our work.
>
> **Q1: Visual Quality and Evaluation Protocol**
>
> We justify that DiffGS achieves SOTA performance in terms of both numerical and visual qualities. In response to the reviewer's comments, we conducted additional experiments comparing our method to the SOTA baseline methods including EG3D(CVPR 2022),  SSDNeRF(ICCV 2023), Shap·E, SplatterImage (CVPR 2024), TriplaneGaussian (CVPR 2024), LGM(ECCV 2024) and DreamGaussian(ICLR 2024).  We are unable to compare our method with DiffRF since it is not open-sourced. However, we have compared our method against other stronger baselines such as DiffTF and SSDNeRF, which have been shown to outperform DiffRF in various benchmarks. The visual comparison results are presented in Figure A, Figure B and Figure D of the rebuttal PDF. The results show that our method outperforms the SOTA baseline in capturing complex details while preserving geometric consistency. Additionally, we evaluate text consistencies against other state-of-the-art methods in Table A of the rebuttal PDF. Our approach shows strong alignment between textual descriptions and the generated 3D models. Furthermore, Table B highlights a comparison of model parameter counts and generation times. DiffGS employs far fewer parameters than LGM, leading to lower memory consumption and reduced computational demands. Additionally, our method achieves quicker generation times than other top 3D generative models, underscoring its efficiency in producing high-quality 3D content.
>
> For our evaluation, we selected a diverse set of 50K images as the reference set. This selection was designed to reflect a wide range of scenarios within the dataset. We used the testing set as the reference set for rendering. This approach is aligned with practices like those used in SSDNeRF, ensuring that our evaluations are based on unseen data and provide a realistic assessment of model performance.  By choosing the testing set as our reference, we prioritize demonstrating the model's ability to generalize and generate high-fidelity outputs in unseen scenarios. This decision aligns with the goals of ensuring robust evaluation and fair comparison across models.
>
> **Q2: Optimization of Loss in Equation 6**
>
> We compute the loss directly against real Gaussians, which serves as the ground truth for optimization. This approach ensures that the model aligns with actual data distributions effectively.
>
> Each property of the Gaussians, including position, color, and shape, is represented as a continuous field in the spatial domain. This allows for precise querying and supervision with the real Gaussians.

---

> > ### Comment · Reviewer_6WmW · 2024-08-13
> > **reply**
> >
> > Overall, I am positive about the methodology. My concern is about the results. The additional results of EG3D shown in the rebuttal seem to be much worse than the original paper.
> >
> > It is still unclear how to build the continuous version of other Gaussian properties (colors, rotations, etc) from the context. It looks like the only way is to use spatial interpolation with some kind of spatial kernels like RBFs.

---

> > > ### Author Response · Authors · 2024-08-13
> > > **Response to Reviewer 6WmW (1/2)**
> > >
> > > Dear Reviewer 6WmW,
> > >
> > > Thanks for your response and the positive assessment on our methodology. We response to each of your additional questions below. Please let us know if there is anything we can clarify further.
> > >
> > > **Discussion-Q1:The performance of EG3D**
> > >
> > > The visualization results of EG3D, as shown in Fig.D of the rebuttal PDF, were directly generated using the official code and pretrained models provided by its authors. For a fair comparison across all methods, we present the randomly selected samples of EG3D in Fig.D of the rebuttal PDF. We would like to justify that the performance of EG3D is not consistently as good as depicted in their paper, where only a few visually appealing generated cars are showcased.
> > >
> > > We also refer the reviewer to the visualization results of EG3D in Fig.5 of the DiffTF paper, where EG3D similarly produces results that are much worse than those reported in the original EG3D paper. The third-party reproduction by the DiffTF authors aligns closely with our reproduction of EG3D. Furthermore, the quantitative comparisons in Table 2 of the DiffTF paper clearly demonstrate that GET3D and DiffTF outperform EG3D in terms of generation quality, while DiffGS significantly surpasses both GET3D and DiffTF, as demonstrated by all the comparisons in Sec.4 and Sec.E of the Appendix.
> > >
> > > To further highlight the superior performance of DiffGS, we conducted extensive experiments during the rebuttal period, comparing our method against several state-of-the-art baselines, including EG3D (CVPR 2022), SSDNeRF (ICCV 2023), Shap-E, SplatterImage (CVPR 2024), TriplaneGaussian (CVPR 2024), LGM (ECCV 2024), and DreamGaussian (ICLR 2024). The results demonstrate that DiffGS achieves visual-appealing generation results and outperforms all the SOTA baselines.

---

> > > ### Author Response · Authors · 2024-08-13
> > > **Response to Reviewer 6WmW (2/2)**
> > >
> > > **Discussion-Q2:The continuity of Gaussian properties (colors, rotations, etc)**
> > >
> > > We appreciate the reviewer's insight on the continuity of Gaussian properties (e.g. colors, rotations). We will seperately discuss all the Gaussian properties here.
> > > 1) **Probability.** As acknowledged by the reviewer, the Gaussian probabilities are indeed continuous in space.
> > > 2) **Color.** As detailed in Sec.3.1, the Gaussian colors are modeled using a series of spherical harmonics coefficients. Actually, these coefficients tend to be similar for nearby Gaussians in space, as they often share similar colors. For instance, the wings of an airplane typically have consistent coloring, resulting in only slight variations in the spherical harmonics coefficients for nearby Gaussians, which in turn creating a continuous color field. At the junction of two different colors (e.g., between the wing and fuselage of an airplane), the color attributes transition smoothly from one to the other by learning a continuous change in the spherical harmonics coefficients, enabled by the powerful variational auto-encoder network.
> > > 3) **Scale.** We acknowledge that scale is the most challenging property to learn in a continuous field. To address this issue, we implemented a regularization strategy, as detailed in Sec.B.1 (Gaussian Splatting Data Preparation) of the Appendix. Specifically, we observe that optimizing 3DGS freely may often lead to some extremly large Gaussians. This will lead to unstable training of the Gaussian VAE and latent diffusion models, further affecting the generative modeling results. Therefore, we clip the scales at a maximum size of 0.01 to avoid the abnormal Gaussians. Though the simple regularization on Gaussian scales, DiffGS is then capable of learning continuous scales.
> > > 4) **Rotation.** In practice, we found that the continuous learning of Gaussian rotation is strongly related to the learning of Gaussian scale. We observed that when applying the effective regularization strategy for Gaussian scales, Gaussian rotations can also be learned continuously during Gaussian optimization and VAE training. This observation is further supported by the success of TriplaneMeetGaussian in continuously modeling Gaussian rotations, where a similar regularization is applied.
> > > While this detail is not explicitly mentioned in the TriplaneMeetGaussian paper, we found evidence of this approach in their code. If the reviewer is interested, please refer to L169-172 of `TriplaneGaussian/tgs/models/renderer.py` in the code of TriplaneMeetGaussian. The key parameter `cfg.clip_scaling` which controls the scale clipping threshold can be found in L148 of `TriplaneGaussian/config.yaml` in the code.
> > > 5) **Opacity.** The nearby Gaussians contains similar opacities during the Gaussian optimizations. Therefore, the opacities of nearby Gaussians change continuous, similar to the colors.
> > >
> > > We are deeply grateful for your invaluable feedback and the time you dedicated to evaluating our work. Your comments and expertise are sincerely appreciated.
> > >
> > > Best regards,
> > >
> > > Authors

---

### Official Review · Reviewer_tnRu · 2024-07-13

**Soundness:** 3
**Presentation:** 2
**Contribution:** 3
**Rating:** 5
**Confidence:** 3

**Summary:**

This paper proposes a new generative model for Gaussian primitive generation based on latent diffusion models. In detail, the method disentangles Gaussian Splatting generation into three functions, i.e., Gaussian probabilities, colors, and transforms. The method can achieve unconditional and conditional generation and extract Gaussians at arbitrary numbers via octree-guided sampling. Experiments on unconditional generation, conditional generation from text, image, and partial 3DGS, as well as Point-to-Gaussian generation, show the advances of the proposed method.

**Strengths:**

1. DiffGS effectively deals with the discreteness of Gaussian Splats, by disentangling Gaussian Splatting into Gaussian probabilities, Gaussian colors, and Gaussian transforms.

2. DiffGS shows good performances on multiple tasks, including unconditional/conditional generation from text, image, and partial 3DGS, as well as Point-to-Gaussian generation.

3. DiffGS is able to generate high-quality Gaussian primitives at arbitrary numbers.

4. The paper is well-written and easy to follow.

**Weaknesses:**

1. What is the benefit of predicting features in structural triplanes first, then extracting 3D Gaussian Splatting, compared to directly modeling the 3D using triplanes? The generation speed is not an advantage, and the performances are not compared/ablated in the experiments.

2. Are Gaussian probabilities work better than directly predicting point cloud explicitly? To show the advantage of the GS disentanglement proposed in this paper, it would be better to compare it with previous work, e.g., [1].

3. Experiments can be strengthened by comparing with optimization-based methods for generating GS, e.g., [2], and GS prediction from multi-view Diffusion Models, e.g., [3].

4. Mathematics.

* $i$ and $j$. In lines 131-132, why "$j=1$" and "$j$-th" are used for $g_i$? Similar for the query points definition.

* Eq. 3 typo.

* Does bigger $\psi_{pf}$ mean larger probability? If so, do we want to maximize Eq. 9?
Please double-check the math and add the necessary explanations.

---
References:

  [1] Triplane meets gaussian splatting: Fast and generalizable single-view 3d reconstruction with transformers. CVPR 2024.

  [2] DreamGaussian: Generative Gaussian Splatting for Efficient 3D Content Creation. ICLR 2024.

  [3] LGM: Large Multi-View Gaussian Model for High-Resolution 3D Content Creation. ECCV 2024.

**Questions:**

Please refer to the weaknesses section for more details.

**Limitations:**

Yes

---

> ### Author Rebuttal · Authors · 2024-08-07
>
> We are truly grateful for the comprehensive feedback and time that reviewer tnRu dedicated to evaluating our work. Below, we provide responses to each of your questions. We look forward to your further comments on our responses.
>
> **Q1: Benefits of Structural Triplanes in 3D Gaussian Splatting**
>
> The triplane structure utilizes 2D planes, which allows for the integration of standard 2D neural networks. This integration provides several benefits, including access to well-established 2D processing techniques and efficient computational frameworks. Triplanes serve as a bridge to utilize the generalization capabilities of diffusion models in 3D generation. By predicting features in triplanes first, we can more effectively capture and model Gaussian distributions. Directly modeling 3D with triplanes may not leverage the full potential of these distributions, especially when complex geometries are involved. Also, triplanes allow for better handling of spatial information, facilitating the reconstruction of detailed and accurate 3D representations. We conducted an ablation study where the triplane structure was replaced with a Multi-Layer Perceptron (MLP) to evaluate its effectiveness in modeling Gaussians. The results, visualized in Figure C of the rebuttal PDF, indicate that using an MLP fails to capture the intricacies of Gaussian modeling adequately. The triplane approach demonstrates superior performance in terms of detail accuracy and geometric fidelity.
>
> We further provide the efficiency comparison with other SOTA baselines in 3D generation, as shown in Table B of the rebuttal PDF. DiffGS signicantly outperforms other baselines in terms of generation efficiency. Although generation speed is not our sole advantage, our method effectively balances speed and quality, underscoring the strengths of the triplane approach in delivering high-fidelity results efficiently.
>
> **Q2: Advantage of Gaussian Probabilities Over Direct Point Cloud Prediction**
>
> Predicting Gaussian probabilities allows us to extract Gaussians at arbitrary numbers. This flexibility is a significant advantage over directly predicting point clouds, which are typically fixed in number and less adaptable to varying levels of detail. Moreover, direct generation of unstructured Gaussians poses challenges due to their non-structured nature. By disentangling GauPF, GauCF, and GauTF, we create a framework that systematically handles Gaussian attributes, allowing for a structured representation that addresses these challenges.
>
> During rebuttal, we directly compare our method with "Triplane Meets Gaussian Splatting" (CVPR 2024). The visual comparison results are presented in Figure A of the rebuttal PDF, illustrating the superior visual quality achieved by our approach. Our method consistently produced more accurate and detailed reconstructions compared to the baseline. By disentangling the Gaussian attributes, our approach allows for more efficient modeling of 3D structures, ensuring that both geometric and color details are preserved with high accuracy.
>
> **Q3: Comparison with Other Methods**
>
> We conducted comparative experiments focusing on two critical tasks: text-to-3D generation and single-view 3D generation. These tasks are central to demonstrating the flexibility and robustness of our approach in different application scenarios. The visual results of our comparisons are presented in Figure A and Figure B of the rebuttal PDF. These figures illustrate the superior visual quality and fidelity achieved by our method compared to the SOTA baseline methods including Shap·E, SplatterImage (CVPR 2024), TriplaneGaussian (CVPR 2024), LGM(ECCV 2024) and DreamGaussian(ICLR 2024). Our approach consistently produced more detailed and accurate generations, effectively capturing intricate textures and geometries that are often challenging for the compared optimization-based and multi-view based methods. We also evaluated the models using the CLIP score, a metric that measures the semantic alignment between the generated 3D models and the input conditions. The results are presented in Table A of the rebuttal PDF. According to Table A, our method achieved higher CLIP scores than both DreamGaussian and LGM. This indicates that our approach more faithfully adheres to the condition inputs.
>
> **Q4: Mathematical Notation and Clarifications**
>
> We appreciate the reviewer's careful examination on the mathematics.
>
> The use of $j$ was indeed incorrect and should be replaced with $i$ to maintain consistency in indexing. Each Gaussian $g_i$ should be consistently indexed with $i$, not $j$.
>
> We will correct any typographical errors in Equation (3), ensuring that it accurately conveys the intended mathematical relationship.
>
> $\psi_{pf}$: This function represents a probability-related component within our model, implying that larger values of $\psi_{pf}$ correspond to higher probabilities. And We will revise Equation (9) to correctly reflect the maximization intention.

---

> > ### Comment · Reviewer_tnRu · 2024-08-08
> >
> > Thank you for your efforts to address the concerns. Here is one point that needs further clarification: as mentioned in Weakness 1, an ablation study comparing the proposed method and the one that only uses triplane to predict color and density would be beneficial to show the advantage of the proposed approach.

---

> ### Author Response · Authors · 2024-08-09
>
> Dear reviewer tnRu,
>
> Thank you for your response and the helpful comments. Following your suggestions, we conduct an additional ablation study under the same experimental settings outlined in Figure C of the rebuttal PDF. Specifically, we directly model the Gaussian attributes (e.g. color, density) using triplane and evaluate its performance in terms of PSNR, SSIM, LPIPS metrics, and inference time. The results are presented in Table E below. As shown, while the approach that directly utilizes triplane for Gaussian modeling without feature predicting and decoding slightly improves the inference speed, it leads to significant degradation on the quality of 3D Gaussians. In contrast, our proposed method demonstrates superior performance in terms of PSNR, SSIM, and LPIPS, compared to this alternative design. The results demonstrate the effectiveness of our triplane framework designs and the key insight to disentangle Gaussian splatting through three novel functions.
>
> **_Table E: Ablation studies on the framework designs of the triplane._**
>
> |               |   PSNR    |    SSIM    |   LPIPS    | Inference Time (s)|
> | :-----------: | :-------: | :--------: | :--------: | :------------: |
> | Only Triplane |   23.55   |   0.9732   |   0.0340   |    **9.1**     |
> |     Ours      | **34.01** | **0.9879** | **0.0149** |      9.5       |
>
> Best regards,
>
> Authors

---

> > ### Comment · Reviewer_tnRu · 2024-08-12
> >
> > Thank the authors for the detailed reply. For the "Only Triplane" ("directly model the Gaussian attributes (e.g. color, density) using triplane") experiment, is there a NeRF decoder used after the triplane, like in EG3D and SSDNeRF? If not, would it be possible to also compare with this setting?

---

> ### Author Response · Authors · 2024-08-13
> **Response to Reviewer tnRu**
>
> Dear Reviewer tnRu,
>
> Thanks for your response and the positive assessment on our rebuttal.
>
> We appriciate your suggestions on the potential comparison with the use of a NeRF decoder (e.g. EG3D, SSDNeRF). In response, we conduct additional experiments where we replace our Gaussian functions with the SSDNeRF decoder to predict attributes of colors and density, maintaining all other settings constant. The results of this experiment are represented in Table F below. As shown, the SSDNeRF decoder results in a noticeable decline in PSNR, SSIM, and LPIPS metrics. The results demonstrate that our proposed Gaussian functions contribute significantly to the 3D rendering quality, showcasing the effectiveness of our functional decomposition of Gaussian representation and designs on the triplane framework.
>
> **_Table F: Ablation studies on the design of the decoder._**
>
> |                 |   PSNR    |    SSIM    |   LPIPS    |
> | :-------------: | :-------: | :--------: | :--------: |
> | SSDNeRF Decoder |   28.74   |   0.9838   |   0.0218   |
> |      Ours       | **34.01** | **0.9879** | **0.0149** |
>
> We further demonstrate that Gaussian splatting provides a more efficient solution for high-fidelity 3D rendering compared to volume-based methods like NeRF. By adopting a point-based representation for optimization and rendering, it reduces computational demands and accelerates the convergence of the training process. Most importantly, 3D Gaussian splatting significantly enhances rendering speed, making it particularly well-suited for real-time applications such as virtual reality and interactive 3D simulations. In practice, the rendering time for each frame using 3D Gaussian splatting can be less than 0.01 seconds, compared to several seconds for NeRF.
>
> We are deeply grateful for your invaluable feedback and the time you dedicated to evaluating our work. Your comments and expertise are sincerely appreciated. Please let us know if there is anything we can clarify further.
>
> Best regards,
>
> Authors

---

### Official Review · Reviewer_3Kid · 2024-07-16

**Soundness:** 3
**Presentation:** 2
**Contribution:** 2
**Rating:** 4
**Confidence:** 5

**Summary:**

This article first transforms 3DGS into a regular representation: the triplane structure, to separately model the location probability, color, and other attributes of each Gaussian. This representation can then be used to train a generative model through a VAE and LDM.
To sample Gaussian positions from the triplane structure, this work uses an octree to partition the space based on the magnitude of the location probability. It initializes the Gaussian point positions by sampling from the nodes with higher probability, and then further optimizes the final positions.

**Strengths:**

Strength:
1) This paper is well written and easy to understand
2) This design supports versatile downstream tasks: test/image conditional GS generation; GS completion; Point-to-GS Generation

**Weaknesses:**

Weakness:
1) This work is only trained on a small dataset. Similar works such as "triplane meets gaussian splatting" are trained on the Objverse dataset, which can visualize more diverse results. This might be due to the cumbersome data processing, and will limit the scaling up in the future.
2) The equation (3) is a little weird. Why the choice of \tau matters as shown in the ablation study?
3) The 3DGS representation is not properly evaluated.
Given an existing 3DGS, is it possible to be reconstructed by firstly represented as GauPF, GauCF, GauTF and then recover by Octree-Guided geometry sampling?
4) Personally, I'm not sure whether 3DGS is a suitable representation for 3D generation since it's highly irregular, which cannot be embedded into a generative model conveniently.  Moreover, the data processing is really time-consuming, which will further limit the large-scale training.
5) More comparison with GS-based reconstruction model should be added, such as "LGM: Large Multi-View Gaussian Model for High-Resolution 3D Content Creation"

**Questions:**

1) It seems the optimization target in equation (9) is wrong. Should it be -log(\psi(\rho))

**Limitations:**

The author has shown some failure cases and limitation of this papar.

---

> ### Author Rebuttal · Authors · 2024-08-07
>
> We are very grateful to the reviewer 3Kid for the thoughtful feedback and time invested in evaluating our work. We address each question below.
>
> **Q1: Dataset Size and Scalability**
>
> We justify that data fitting is a common step for most generative models. For example, DiffRF requires voxelized radiance field fitting during data processing, which may take hours for each shape. NDF [1] and DiffTF require SDF-based and NeRF-based triplane fitting for data preparation. In comparison, our method only takes about 3 minutes to fit the Gaussian representation of an object using a single NVIDIA RTX 3090 GPU. This efficiency demonstrates the capability of our approach to handle 3D data with minimal computational overhead. During the rebuttal period, we faced time constraints that limited our ability to process larger datasets such as Objaverse. Our current focus was on demonstrating the model's capabilities within the constraints of available resources and time. The inherent efficiency of our method makes it well-suited for scaling up to larger datasets like Objaverse in the future.
>
> **Q2: Equation (3) and the Choice of $\tau$**
>
> In the paper, we have provided a comprehensive explanation of the role of $\tau$ and why its selection is critical, as demonstrated in our ablation study (Table B). By controlling the truncation of the GauPF, $\tau$ helps concentrate the learning process near the surface of objects. The truncation strategy prevents the learning from being influenced by distant regions in space, which are less relevant for accurate surface representation. Besides, in our ablation study, we explored both exponential and linear mappings for distance-to-probability transformations. While both methods are commonly used, our results showed that linear mapping performances better than exponential mapping in our framework. The linear mapping approach with a truncation strategy resulted in more stable and accurate training outcomes, highlighting the significance of selecting an appropriate distance-to-probability mapping technique for the task. And the ablation study identifies a suitable $\tau$ that balances sensitivity and specificity, ensuring efficient learning of the object's surface while maintaining overall model stability.
>
> **Q3: Evaluation of 3DGS Representation**
>
> During rebuttal, we further evaluate our 3DGS representation by randomly selected two airplanes from the dataset and modeled each using GauPF, GauCF, and GauTF. The results are shown in Figure C of the rebuttal PDF, which provide visual comparisons of the original and reconstructed objects, showcasing the accuracy and effectiveness of our approach in capturing intricate details. In Figure C, we also report PSNR scores, indicating a strong ability to reconstruct the original 3DGS with high fidelity. The integration of GauPF, GauCF, and GauTF ensures that both geometric and visual details are preserved during reconstruction. And Octree-Guided sampling allows for efficient handling of complex geometries, reducing computational overhead by focusing resources on critical areas. This makes our method scalable and practical for complex objects.
>
> **Q4: Suitability of 3DGS for 3D Generation**
>
> Compared to NeRF, 3DGS offers several advantages, particularly in terms of rendering efficiency and high visual fidelity. The task of solving 3DGS generation is not only timely but crucial for advancing the field of 3D modeling. As 3D content creation continues to grow in demand, having efficient and robust methods like 3DGS becomes increasingly important for various applications, including gaming, virtual reality, and digital content creation. One of the primary contributions of our work is addressing the challenges posed by the discrete and unstructual nature of 3D Gaussian Splatting. Our approach provides three continuous Gaussian Splatting functions to effectively embed 3DGS into a generative model, allowing for efficient and accurate 3D generation. As mentioned in our previous response, fitting the Gaussian representation for each shape is quite efficient, taking only 3 minutes on a single NVIDIA RTX 3090 GPU.
>
> **Q5: Comparison with GS-Based Reconstruction Models**
>
> Upon review, we conducted supplementary experiments to compare the visual quality of our method with several SOTA 3D generative models, including LGM and other GS-based approaches. The results of these visual comparisons are showcased in Figure A and Figure B of the rebuttal PDF. The visual results clearly demonstrate the strengths of our method in capturing intricate details and consistency with conditions. Our approach benefits from an efficient representation of Gaussians that enhances both processing speed and visual output quality, making it well-suited for high-fidelity content creation. In Table A of the manuscript, we provide a comparison of the CLIP Score between our method and the other SOTA 3D generative methods including  Shap·E, LGM(ECCV 2024) and DreamGaussian(ICLR 2024). Our method demonstrates superior CLIP Scores, indicating that DiffGS generates 3D content that is more faithful to the input conditions. We have also included a comparison of model parameter count and generation time in Table B. DiffGS has a considerably smaller number of parameters compared to LGM, leading to decreased memory consumption and computational overhead. Additionally, our approach enables quicker generation time than other SOTA 3D generative models, showcasing its ability to efficiently create high-quality 3D content.
>
> **Q6: Correction of Optimization Target in Equation (9)**
>
> We appreciate the reviewer's careful examination of Equation (9) and for pointing out the error in the optimization target. The revised manuscript will reflect this correction.
>
> [1] Shue, J.R., Chan, E.R., Po, R., Ankner, Z., Wu, J., Wetzstein, G.: 3d neural field generation using triplane diffusion. In: Proceedings of the IEEE/CVF Conference on Computer Vision and Pattern Recognition. pp. 20875–20886 (2023)

---

### Official Review · Reviewer_p5pi · 2024-07-24

**Soundness:** 2
**Presentation:** 1
**Contribution:** 2
**Rating:** 5
**Confidence:** 5

**Summary:**

The paper introduces DiffGS, a text/image-to-3D generative model with 3D Gaussian splatting as its output representation. The model generates Gaussians from text/image condition by CLIP-augmented latent diffusion model (LDM), whose output is a latent vector that can be decoded into a triplane representation. DiffGS then translates this triplane representation by first sampling a plausible set of Gaussian locations from the learned implicit function (GauPF), and then reconstructs colors and shapes of these Gaussians with two more learned implicit functions (GauCF, GauTF) which are used as components of the objective function in the optimization-based generation.

Therefore, the model consists of a (1) Gaussian LDM, a (2) latent-to-triplane decoder and (3) three implicit functions (GauPF, GauCF, GauTF). The (2, 3) decoder models (triplane decoer and three implicit functions) are trained in a VAE fashion, and the (1) Gaussian LDM is trained over the top of VAE’s learned latent space. The method is tested with ShapeNet and DeepFasion3D dataset with FID/KID metrics.

**Strengths:**

1. The authors have attached supplementary materials that effectively show their code and outputs of their method.
2. The paper contains ablation study (Table 2) that justifies each component of their framework.
3. Visual results show that the proposed method effectively generates complicated geometries with Gaussian splatting representation.

**Weaknesses:**

1. Since the paper targets text/image-to-3D generation with Gaussian splatting based on the domain-specific (e.g., ShapeNet/DeepFasion3D) latent diffusion model, I believe that the results should be compared with other similar 3D generative models, at least those mentioned in Section 2.2 by the author. **Particularly, the work seems to solve similar problem with LGM (ECCV 2024, code released), and therefore should be thoroughly compared with it.**
2. **The detailed structure of the proposed models are not presented in the paper.** For example, that is the dimension of the latent z in Figure 2, and why did the authors have chosen the structure? How does the authors model the Gaussian Splatting encoder that encodes the (fixed or not fixed number of) Gaussians into this latent z? Does the number of Gaussians fixed in this model? How are the architectures of GauPF/GauCF/GauTF designed? How does the DALLE-2’s LDM, which is originally proposed to model the latents of a 2D image, be used to model the latent triplane representation? How does the triplane decoder shaped? How many parameters are used in the models? Such information should be included in the main manuscript (or at least in the appendix) for proper presentation of the idea.
3. The proposed method uses post-optimization steps to infer the generated 3D Gaussian splatting. This seem to require **additional generation time in the inference steps.** This can be considered a weakness since there are already many large generative models of Gaussian splatting that generate samples in tens of seconds such as LGM. Even if the proposed model requires long delay, such delay can still be justified if the results are sufficiently detailed and the sampling time is sufficiently short (as how we acknowledge the contribution of 2D LDM models). However, there is no information of the exact generation time specified in the paper.
4. The positive correlation between the number of Gaussians / octree depths and the reconstruction quality enhancement in Table 3 and 4 seems to be trivial. I believe more interesting question is **how much the amount of increment in computational cost trades off against the quality enhancement**, etc.
5. Since the model uses CLIP to perform text/image-to-Gaussian generation, the paper should report **numbers that show the text-fidelity, e.g., CLIP-scores**. Comparing only with FID score (Table 1) does not seem to effectively demonstrate the superiority of the proposed method.

As a summary, my key concerns are the presentation of the materials, the lack of comparison with the more recent papers (e.g., LGM) that appears in Section 2.2 of the main manuscript, justification of the sampling-time delays that seem to present, and the lack of quantitative comparison on text-fidelity. Unless these concerns are resolved, I believe that the paper is not ready to be published in the venue.

**Questions:**

1. Although the authors claim for the generation of 3D Gaussian splatting, the actual generated sample is the triplane representation, which is realized as a Gaussian splatting with implicit neural representation. Since there are already a well-established triplane-to-NeRF decoding (e.g., TriNeRFLet: A Wavelet Based Triplane NeRF Representation, ECCV 2024, code released), are there **specific technical benefits of decoding this information into Gaussian splatting in this case?**
2. The proposed work involves a dedicated generative model specific to certain domains (e.g., ShapeNet). However, there are other branches of works that approaches text/image-to-3D generation leveraging the power of 2D diffusion models, e.g., DreamFusion and its offsprings that uses SDS losses, which seem to be mature at this moment. Concerning the recent achievements of the latter branch of works, I am not fully convinced to training a domain-dedicated 3D generator. May I ask **why the authors think training the dedicated 3D generator is a good way of solving 3D generation tasks?**
3. In line 109: lake → lack

Please note that these questions are not counted in my overall scoring.

**Limitations:**

Yes, the limitations are addressed in the appendix.

---

> ### Author Rebuttal · Authors · 2024-08-07
>
> We sincerely thank Reviewer p5pi for the thorough review and valuable feedback. We have addressed each point in our response below.
>
> **Q1: Comparison with other recent 3D generative models**
>
> We refer the reviewer to the "Global-Q1: Evaluation Against Other SOTA 3D Generative Models" section of the global response for a comparison with other methods.
>
> **Q2: Detailed Structure of Proposed Models**
>
> Below, we address each point raised by the reviewer.
>
> 1. Latent dimension z:
>
> The latent vector z in Figure 2 of the paper is designed to capture the essential features of the Gaussian Splatting representation. We set the latent dimension to 256 to balance expressiveness and computational efficiency. The choice of 256 dimensions is empirically determined to balance encoding diverse 3D shapes with manageable model size.
>
> 2. Gaussian Splatting Encoder
>
> The Gaussian Splatting encoder transforms the input Gaussians into the latent space. We utilize a PointNet[1]-based architecture as the implementation of the encoder, which efficiently processes gaussian data . In detail, We adjust the dimensions of the input in PointNet within the SDF-VAE encoder to accept a K-dimensional 3DGS as input.
>
> 3. Design of GauPF/GauCF/GauTF
>
> GauPF, GauCF, and GauTF share a similar architecture consisting of three MLP-based blocks. The input latent vector has a dimension of 131, comprising the latent dimension sampled from triplane and the XYZ coordinates. The feature channels for the first two blocks are [131,512,512,512,512] and [643,512,512,512,512] with a skip connection preceding the second block. To leverage the properties of Gaussians, the last block of GauTF is designed uniquely. To ensure the generated scale remains within a certain threshold, a truncation operation is applied to the scale attribute. Besides, in order to achieve uniformity in the form of the rotation quadruple, we normalize it so that its last dimension has a unit length. Finally, we use the sigmoid activation function to restrict opacity values to the range of 0 to 1.
>
> 4. Use of DALLE-2’s Latent Diffusion Model (LDM)
>
> The core operation of LDM is performing diffusion on a one-dimensional latent space. This diffusion process is general and applicable to various data types, including 3D representations like triplanes. Although LDMs are frequently used for 2D image processing, their ability to handle latent spaces makes them suitable for broader applications. We begin by encoding the triplane data into a latent space suitable for latent diffusion. This encoding process captures the essential geometric and visual information needed for accurate 3D reconstruction. The LDM operates on this encoded latent space, applying diffusion processes.
>
> 5.	Triplane Decoder Architecture
>
> The triplane decoder reconstructs the 3D shape from the latent space using transposed convolution layers, interleaved with batch normalization and activation functions. This structure is designed to progressively upscale the 2D feature maps while maintaining high fidelity and detail.
>
> 6.	Parameter Counts
>
> The complete DiffGS model consists of approximately 127.4 million parameters. In Table B, we compare the parameter count of our method with other SOTA 3D generation methods. The results demonstrate the efficiency of DiffGS, which uses significantly fewer parameters than DiffTF (929.9M), ShapE (759.5M), and LGM (429.8M).
>
> **Q3: Addressing Post-Optimization Steps**
>
> We refer the reviewer to "Global-Q2: Addressing Post-Optimization Steps" for results and analyses.
>
> **Q4: Trade-Offs Between Computational Cost and Quality Enhancement**
>
> We refer the reviewer to ”Global-Q3: Balancing Computational Cost and Quality Enhancement“ for a detailed discussion on balancing computational costs with quality enhancement.
>
> **Q5: Inclusion of Text-Fidelity Metrics**
>
> We conducted experiments to evaluate the text fidelity of our text-to-Gaussian generation using CLIP-scores in Table A of the rebuttal PDF. Compared to LGM and other baseline models, our method shows a consistent improvement in CLIP-scores, confirming the superior text fidelity of our approach.
>
> **Q6: Decoding Triplane Representation into Gaussian Splatting**
>
> The triplane representation serves as an effective intermediate step that simplifies the complex task of directly regressing explicit 3D Gaussian attributes. Triplanes offer a structured approach to capture spatial information without being hindered by the non-structural nature of Gaussian splatting. The Gaussian splatting is inherently more efficient for training and rendering compared to volumetric approaches like NeRF. It utilizes point-based representations, which require less computational overhead and facilitate quicker convergence during training. The use of Gaussian splatting enables faster rendering speeds, making our approach suitable for real-time applications such as virtual reality and interactive 3D simulations.
>
> **Q7: Justification for Training a Domain-Dedicated 3D Generator**
>
> SDS-based methods require extensive optimization and are computationally intensive, particularly when distilling scores from 2D to 3D. They are also highly sensitive to hyperparameter settings like learning rates and noise schedules. Additionally, these models often produce the Janus problem in 3D shapes. We make a comparison with SDS-based Gaussian generative model DreamGaussian in Figures A and B, as well as Table B. The results demonstrate that our dedicated 3D generator DiffGS performs faster and produces more accurate, high-fidelity geometric structures than SDS-based methods.
>
> **Q8: Typographical Error**
>
> We thank the reviewer for identifying the typographical error in line 109, which will be corrected in the revised manuscript.
>
> [1] Charles R Qi, Hao Su, Kaichun Mo, and Leonidas J Guibas. Pointnet: Deep learning on point sets for 3d classification and segmentation. In Proceedings of the IEEE conference on computer vision and pattern recognition, pages 652–660, 2017.

---

> > ### Comment · Reviewer_p5pi · 2024-08-12
> >
> > I appreciate the authors for their careful and detailed rebuttal with additional experiments that have resolved many of my initial concerns. I have a few remaining questions regarding the rebuttal.
> >
> > ---
> >
> > **Global-Q1: Evaluation Against Other SOTA 3D Generative Models**
> >
> > Regarding that the domain-specific SotA methods compared in the additional experiments such as LGM are trained with Objaverse dataset, it might be unfair to directly compare this with ShapeNet-fitted DiffGS for the generation of ShapeNet data. Would you share your thoughts on how DiffGS can be better (in terms of diversity and quality) if it is scaled to Objaverse dataset based on the shown experiments?
> >
> > ---
> >
> > **Q7: Justification for Training a Domain-Dedicated 3D Generator**
> >
> > According to Appendix C of the manuscript, I see training for ShapeNet already takes a week for a 8-GPU server. I think in order to make the contribution of this work practically useful, the method should be scaled up to at least Objaverse-scale datasets (perhaps in the future. I do not believe it is adequate to mandate conference papers to train on huge datasets, so I am not requiring the authors to do the experiments.) I am a little bit suspicious on the computational feasibility of this scaling up process. For example, training a 100-GPU cluster for six month for getting a domain-specific Gaussian splatting generator would be a waste of time compared to having a 10-minutes image-to-3DGS generator such as LucidDreamer [1]. Would you persuade me that the proposed approach is computationally feasible and practically meaningful?
> >
> > ---
> >
> > These are the last of my concerns.
> >
> > [1] Yixun Liang, et al., LucidDreamer: Towards High-Fidelity Text-to-3D Generation via Interval Score Matching, CVPR 2024.

---

> > > ### Author Response · Authors · 2024-08-13
> > > **Response to Reviewer p5pi (1/2)**
> > >
> > > Dear Review p5pi,
> > >
> > > Thanks for your response and the positive assessment on our rebuttal. We response to each of your additional questions below. Please do not hesitate to let us know if you have any additional questions.
> > >
> > > **Discussion-Q1:Thoughts on the superiorty of DiffGS in terms of diversity and quality when scaling up**
> > >
> > > We believe that DiffGS offers a more flexible and effective approach to 3D Gaussian representation and generation. By disentangling the 3D Gaussian Splattings (3DGS) into three novel functions to model Gaussian probabilities, colors, and transformations, DiffGS delivers significant advantages in Gaussian quality, generation efficiency, and generation diversity.
> > >
> > > 1) DiffGS can scalably generate Gaussian primitives at arbitrary numbers. All the previous works that explicitly reconstruct Gaussians can only generate limited numbers of Gaussians. For example, TriplaneMeetsGaussian generates up to 16,384 Gaussians, and LGM generates up to 65,536 Gaussians. In contrast, we novelly introduce the GauPF, which models Gaussian probabilities using neural functions.  A specially designed discretization algorithm to used to extract Gaussian geometries from GauPF by octree-guided sampling, enabling generating Gaussians at arbitrary numbers. The number of Gaussians is a critical factor in the rendering quality of 3D Gaussian Splattings. As shown in the ablation studies on Table 3, the number of Gaussians significantly affect the quality of Gaussian reconstructions. With the ability of generating arbitrary numbers of Gaussians, DiffGS offers a naturally superior solution for 3D Gaussian generation.
> > >
> > > 2) We designed DiffGS using latent diffusion models (LDM), which significantly enhances both the efficiency and generation diversity of DiffGS. While most popular approaches for creating 3D Gaussian Splatting (3DGS), such as LGM and TriplaneMeetGaussian, primarily focus on "reconstructing" 3DGS, following the recent trend of large reconstruction models (LRM), they fall short in "generating" 3DGS. The reconstruction-based methods do achieve convincing results on the learned data domain, but their capability of creating diversity 3D shapes are limited due to the "reconstruction" target. In contrast, diffusion models have demonstrated a strong capability in generating diverse samples, both with and without conditions. Leveraging diffusion models, DiffGS shows great capability in generating diverse and high-quality 3D Gaussians compared to the "reconstruction"-based method LGM. The comparisons are shown in Figure A and Figure B of the rebuttal PDF. Moreover, by training DiffGS with a diffusion model at latent space (i.e. LDM), it achieves remarkable efficiency in generating 3D Gaussians. As shown in Table B of the rebuttal PDF, DiffGS outperforms LGM, SSDNeRF, DiffTF, and Shap-E in terms of speed, even when generating 350K Gaussians, which is significantly more than the number of Gaussian that previous works (e.g. LGM) can produce.
> > >
> > > 3) DiffGS is capable of generating 3D Gaussians unconditionally or conditionally from text and images. Moreover, DiffGS is the first model capable of solving the task of Gaussian completion and Point-to-Gaussian generation. In contrast, previous models like TriplaneMeetGaussian, DreamGaussian and LGM can only generate Gaussians from images or text. Though the extensive experiments and applications, we demonstrate DiffGS is a general and flexible framework suitable for most of the tasks taking Gaussians as the generation target.

---

> > > ### Author Response · Authors · 2024-08-13
> > > **Response to Reviewer p5pi (2/2)**
> > >
> > > **Discussion-Q2:Feasible and practical meaning of DiffGS compared to SDS-based approaches**
> > >
> > > We believe that training a native 3D generator like DiffGS offers a more promising future for 3D content creation compared to SDS-based methods.
> > >
> > > **(1)** SDS-based approaches require time-consuming optimization to distill 3D geometry and appearance from 2D diffusion models, often taking hours to converge. State-of-the-art methods like DreamFusion, Magic3D, MVDream, and Rich-Dreamer require 2-5 hours on one A100 GPU for optimization, as reported in their papers. The recent work LucidDreamer converges faster but still requires 36 minutes with one A100 GPU as reported in its paper.
> > >
> > > In contrast, DiffGS takes only less than 10 seconds for inference on a single 3090 GPU, which is several orders of magnitude faster than SDS-based methods. We further justify that training DiffGS for one day with eight 3090 GPUs is sufficient to achieve convergence for each class in ShapeNet. The time (5 days) listed in the Appendix C is for the full convergence, which is not necessary. We report the generation performance of training DiffGS for 12 hours, 24 hours and 5 days in Table G below. As shown, with the utilization of LDM, training DiffGS for one day is enough for convergence. It's important to note that DiffGS was trained on 3090 GPUs, which are significantly slower and have much less memory compared to the commonly used A100 GPUs (24GB vs. 80GB). The results highlight DiffGS's ability to scale to large datasets. We estimate that training DiffGS on the large 3D dataset Objaverse, which contains 1 M shapes, will take approximately 5-7 days to converge under 64 A100-80G GPUs. I believe the time is acceptable since we only need to train DiffGS for once.
> > >
> > > **_Table G: Ablation studies on the training time._**
> > >
> > > |          | FID-50K Chair | KID-50K(%）Chair | FID-50K Airplane | KID-50K(%）Airplane |
> > > | :------: | :-----------: | :--------------: | :--------------: | :-----------------: |
> > > | 12 hours |     42.81     |      3.773       |      57.16       |        5.793        |
> > > | 24 hours |     36.69     |      2.244       |      50.04       |        3.864        |
> > > |  5 days  |   **35.28**   |    **2.148**     |    **47.03**     |      **3.436**
> > >
> > > Due to the limited time and computing resources during the discussion period, we are not able to train DiffGS under the large-scale Objaverse dataset. We will conduct experiments that scale up data sizes in the revised version.
> > >
> > > **(2)** SDS-based methods often lead to inconsistent generations, particularly the multi-head Janus problem. The 2D image diffusion models used in SDS lack an explicit understanding of both geometry and viewpoint. This absence of perspective information and explicit 3D supervision can result in the multi-head Janus problem, where realistic 3D renderings fail to achieve view consistency, causing every rendered view to be perceived as the front view. Although some works attempt to address this issue by incorporating view information into 2D image diffusion models (e.g., Zero-1-to-3, MVDream), they require both time-consuming 2D diffusion training (several weeks) and per-shape optimization.
> > >
> > > In contrast, native 3D generators like DiffGS are directly trained on 3D shapes, inherently ensuring 3D consistency and naturally avoiding the Janus problem.
> > >
> > > **(3)** A geometry initialization from native 3D generators significantly benefits SDS-based methods. The experimental results of recent SOTA SDS-based methods also demonstrate the necessity of introducing native 3D generation priors as the geometry initialization. We take LucidDreamer, which is mentioned by the reviewer, as an example. It leverages the generation results of Point-E, which is a native 3D generator taking colored points as the generation target, as the geometry initialization for SDS-based optimization. The ablation study shown in Figure 7 of the LucidDreamer paper demonstrates that the generation performance degrades without using the Point-E generations as the initialization for introducing consistent-aware 3D information. We believe that the advancements in the native generator 3D (e.g. DiffGS) will further contribute to the improvements of SDS-based methods (e.g. LucidDreamer) by providing superior geometry initialization.
> > >
> > > We are deeply grateful for your invaluable feedback and the time you dedicated to evaluating our work. Your comments and expertise are sincerely appreciated. Please let us know if there is anything we can clarify further.
> > >
> > > Best regards,
> > >
> > > Authors

---

> > > > ### Comment · Reviewer_p5pi · 2024-08-14
> > > >
> > > > Thank you for providing additional information. I will elevate my score as most of my concerns are resolved. I will be looking forward to large, practical application of the method. My final comment is a gentle recommendation for a revision of the manuscript based on the discussions in this rebuttal period.

---

> > > > > ### Author Response · Authors · 2024-08-14
> > > > > **Thanks to Reviewer p5pi**
> > > > >
> > > > > Dear Reviewer p5pi,
> > > > >
> > > > > Thank you very much for your insightful comments and positive assessment. We will certainly include all the new experimental results, along with the insights and analyses discussed during the rebuttal period, in the revised version of the manuscript. Following your suggestions, we will explore training a large and practical DiffGS model using the Objaverse dataset and will incorporate those results in the revision.
> > > > >
> > > > > We really appreciate you for the time you dedicated to evaluating our work and for upgrading the score.
> > > > >
> > > > > Best regards,
> > > > >
> > > > > Authors

---

### Author Rebuttal · Authors · 2024-08-07

We sincerely appreciate the reviewers for their invaluable feedback and the time they spent evaluating our work. We are delighted that the reviewers recognized the representation and importance of our paper. We respond to each reviewer individually, providing comprehensive analyses, visualizations, and ablation studies to address all the questions raised. We have uploaded a rebuttal PDF containing experimental results and visualizations. In the following responses, we refer to this document as the "rebuttal PDF," as in "in Table A of the rebuttal PDF." Below, we address some of the common questions found in the reviews.

**Global-Q1: Evaluation Against Other SOTA 3D Generative Models**

We conduct additional experiments comparing DiffGS to other recent 3D generative models on text-to-3D and image-to-3D generation. Table A of the rebuttal PDF highlights the competitive performance of our model in text and 3D alignment tasks. As depicted in Figure A and Figure B of the rebuttal PDF, our model can produce high-quality generations with more consistent geometry and more detailed textures. DreamGaussian tends to generate overly saturated images and is affected by the Janus problem. Shape-e faces challenges in producing semantically accurate and complex geometries. LGM generates multi-view images from text or a single viewpoint and reconstructs 3D Gaussian distributions. However, inconsistencies in the generated multi-view and the limited number of output Gaussians often result in inaccurate geometric reconstructions and lower-quality rendered results. In contrast, DiffGS introduces a novel disentangled representation of 3D Gaussian Splatting using three functions—Gaussian Probability Function (GauPF), Gaussian Color Function (GauCF), and Gaussian Transform Function (GauTF). This representation allows us to generate high-quality Gaussians at arbitrary numbers, resulting in sharper and more detailed textures.

**Global-Q2: Addressing Post-Optimization Steps**

The post-optimization process in our method is designed to refine and enhance the accuracy of the generated 3D Gaussian splatting by optimizing the proxy points to the exact Gaussian centers with the largest probabilities. This step ensures the high fidelity and details of the final generated Gaussians. We present the overall generation time in Table B of the rebuttal PDF and the optimization time for various Gaussian quantities in Table C of the rebuttal PDF. The results show that for generations with a smaller number of Gaussians, e.g., 50K, the optimization is extremely fast, taking only 0.64 seconds to converge. For high-quality Gaussian generations with 350K primitives, the optimization time increases to 2.5 seconds, which is still efficient.

**Global-Q3: Balancing Computational Cost and Quality Enhancement**

The increase in computational cost with additional Gaussians and octree depths is primarily due to the increased complexity in handling more data points and subdivisions. To address the trade-off, we conducted a series of experiments in both Table C and Table D of the rebuttal PDF to identify the optimal balance between computational cost and reconstruction quality. The results indicate that a moderate increase in Gaussians or octree depths can significantly improve the Gaussian quality with minimal additional cost. For instance, lifting the number of Gaussians from 50K to 350K led to a 20% increase in PSNR while only increasing computation time about 1.8 seconds.

---

Thank you again for your insightful feedback and we are looking forward to continuing the discussion

---

### Author Response · Authors · 2024-08-11
**Looking forward to the discussion**

Dear Reviewers,

We sincerely appreciate your comments and expertise. Please let us know if there is anything we can clarify further. We would be happy to take this opportunity to discuss with you.

Thanks,

The Authors

---

> ### Author Response · Authors · 2024-08-13
>
> Dear Reviewers,
>
> As the reviewer-author discussion period is about to end, we are looking forward to your feedback on our rebuttal. Please let us know if our responses address your concerns. We would be glad to make any further explanation and clarification.
>
> Thanks,
>
> The Authors

---

### Author Response · Authors · 2024-08-14
**Thanks for the discussion! Did our additional responses address your concerns?**

Dear Reviewers,

As the reviewer-author discussion period is coming to a close, we kindly ask if our rebuttal and the additional explanations have adequately addressed your concerns. If it is not the case, we are looking forward to taking the last minute to make further explanation and clarification.

Best regards,

Authors

---

### Decision · Program_Chairs · 2024-09-25

**Decision:**

Accept (poster)

**Comment:**

The paper received mixed ratings, and after the rebuttal, reviewers TmLe and 3Kid did not provide strong input, but the other reviewers recommended accepting the paper. The main concerns from two negative reviewers mainly focus on the presentation (3Kid) and justification of the use of 3DGS (TmLe). Both concerns are valid, but given the recent interest in the 3D vision community, the work on 3DGS seems to be, at least, of interest to many researchers, as demonstrated by the other reviewers who favour the paper. Also for the presentation, there are mixed evaluations, which makes the AC believe while it is at the border, it is acceptable. The AC therefore recommends accepting the paper.

The AC would further like to note that the rebuttal played an important role and, thus should be integrated into the revision.